# Munchausen Reinforcement Learning

**Nino Vieillard**
Google Research, Brain Team
Université de Lorraine, CNRS, Inria, IECL
F-54000 Nancy, France
`vieillard@google.com`

**Olivier Pietquin**
Google Research, Brain Team
`pietquin@google.com`

**Matthieu Geist**
Google Research, Brain Team
`mfgeist@google.com`

## Abstract

Bootstrapping is a core mechanism in Reinforcement Learning (RL). Most algorithms, based on temporal differences, replace the true value of a transiting state by their current estimate of this value. Yet, another estimate could be leveraged to bootstrap RL: the current policy. Our core contribution stands in a very simple idea: adding the scaled log-policy to the immediate reward. We show that slightly modifying Deep $Q$-Network (DQN) in that way provides an agent that is competitive with distributional methods on Atari games, without making use of distributional RL, $n$-step returns or prioritized replay. To demonstrate the versatility of this idea, we also use it together with an Implicit Quantile Network (IQN). The resulting agent outperforms Rainbow on Atari, installing a new State of the Art with very little modifications to the original algorithm. To add to this empirical study, we provide strong theoretical insights on what happens under the hood – implicit Kullback-Leibler regularization and increase of the action-gap.

## 1 Introduction

Most Reinforcement Learning (RL) algorithms make use of Temporal Difference (TD) learning [29] in some ways. It is a well-known bootstrapping mechanism that consists in replacing the unknown true value of a transiting state by its current estimate and use it as a target for learning. Yet, agents compute another estimate while learning that could be leveraged to bootstrap RL: their current policy. Indeed, it reflects the agent's hunch about which actions should be executed next and thus, which actions are good. Building upon this observation, our core contribution stands in a very simple idea: optimizing for the *immediate* reward *augmented* by the scaled log-policy of the agent when using any TD scheme. We insist right away that this is different from maximum entropy RL [36], that *subtracts* the scaled log-policy to *all* rewards, and aims at maximizing both the expected return and the expected entropy of the resulting policy. We call this general approach "*Munchausen* Reinforcement Learning" (M-RL), as a reference to a famous passage of *The Surprising Adventures of Baron Munchausen* by Raspe [24], where the Baron pulls himself out of a swamp by pulling on his own hair.

To demonstrate the genericity and the strength of this idea, we introduce it into the most popular RL agent: the seminal Deep $Q$-Network (DQN) [23]. Yet, DQN does not compute stochastic policies, which prevents using log-policies. So, we first introduce a straightforward generalization of DQN to maximum entropy RL [36, 17], and then modify the resulting TD update by adding the scaled log-policy to the immediate reward. The resulting algorithm, referred to as Munchausen-DQN (M-DQN), is thus genuinely a slight modification of DQN. Yet, it comes with strong empirical performances. On the Arcade Learning Environment (ALE) [6], not only it surpasses the original DQN by a large

margin, but it also overtakes C51 [8], the first agent based on distributional RL (distRL). As far as we know, M-DQN is the first agent not using distRL that outperforms a distRL agent[1]. The current state of the art for single agent algorithms is considered to be Rainbow [18], that combines C51 with other enhancements to DQN, and does not rely on massivly distributed computation (unlike R2D2 [19], SEED [12] or Agent57 [4]). To demonstrate the versatility of the M-RL idea, we apply the same recipe to modify Implicit Quantile Network (IQN) [11], a recent distRL agent. The resulting Munchausen-IQN (M-IQN) surpasses Rainbow, installing a new state of the art.

To support these empirical results, we provide strong theoretical insights about what happens under the hood. We rewrite M-DQN under an abstract dynamic programming scheme and show that it implicitly performs Kullback-Leibler (KL) regularization between consecutive policies. M-RL is not the first approach to take advantage of KL regularization [27, 2], but we show that, because this regularization is implicit, it comes with stronger theoretical guarantees. From this, we link M-RL to Conservative Value Iteration (CVI) [20] and Dynamic Policy Programming (DPP) [3] that were not introduced with deep RL implementations. We also draw connections with Advantage Learning (AL) [5, 7] and study the effect of M-RL on the action-gap [13]. While M-RL is not the first scheme to induce an increase of the action-gap [7], it is the first one that allows quantifying this increase.

## 2   Munchausen Reinforcement Learning

RL is usually formalized within the Markov Decision Processes (MDP) framework. An MDP models the environment and is a tuple $\{\mathcal{S}, \mathcal{A}, P, r, \gamma\}$, with $\mathcal{S}$ and $\mathcal{A}$ the state and action spaces, $P$ the Markovian transition kernel, $r$ the bounded reward function and $\gamma$ the discount factor. The RL agent interacts with the MDP using a policy $\pi$, that associates to every state either an action (deterministic policy) or a distribution over actions (stochastic policy). The quality of this interaction is quantified by the expected discounted cumulative return, formalized as the state-action value function, $q_\pi(s, a) = \mathbb{E}_\pi[\sum_{t=0}^{\infty} \gamma^t r(s_t, a_t)|s_0 = s, a_0 = a]$, the expectation being over trajectories induced by the policy $\pi$ and the dynamics $P$. An optimal policy satisfies $\pi_* \in \operatorname{argmax}_\pi q_\pi$. The associated optimal value function $q_* = q_{\pi_*}$ satisfies the Bellman equation $q_*(s, a) = r(s, a) + \gamma E_{s'|s,a}[\max_{a'} q_*(s', a')]$. A deterministic *greedy* policy satisfies $\pi(a|s) = 1$ for $a \in \operatorname{argmax}_{a'} q(s, a')$ and will be written $\pi \in \mathcal{G}(q)$. We also use *softmax* policies, $\pi = \operatorname{sm}(q) \Leftrightarrow \pi(a|s) = \frac{\exp q(s,a)}{\sum_{a'} \exp q(s,a')}$.

A standard RL agent maintains both a $q$-function and a policy (that can be implicit, for example $\pi \in \mathcal{G}(q)$), and it aims at learning an optimal policy. To do so, it often relies on Temporal Difference (TD) updates. To recall the principle of TD learning, we quickly revisit the classical $Q$-learning algorithm [34]. When interacting with the environment the agent observes transitions $(s_t, a_t, r_t, s_{t+1})$. Would the optimal $q$-function $q_*$ be known in the state $s_{t+1}$, the agent could use it as a learning target and build successive estimates as $q(s_t, a_t) \leftarrow q(s_t, a_t) + \eta(r_t + \gamma \max_{a'} q_*(s_{t+1}, a') - q(s_t, a_t))$, using the Bellman equation, $\eta$ being a learning rate. Yet, $q_*$ is unknown, and the agent actually uses its current estimate $q$ instead, which is known as bootstrapping.

We argue that the $q$-function is not the sole quantity that could be used to bootstrap RL. Let's assume that an optimal deterministic policy $\pi_*$ is known. The log-policy is therefore $0$ for optimal actions, and $-\infty$ for sub-optimal ones. This is a very strong learning signal, that we could add to the reward to ease learning, without changing the optimal control. The optimal policy $\pi_*$ being obviously unknown, we replace it by the agent's current estimate $\pi$, and we assume stochastic policies for numerical stability. To sum up, M-RL is a very simple idea, that consists in replacing $r_t$ by $r_t + \alpha \ln \pi(a_t|s_t)$ in any TD scheme, assuming that the current agent's policy $\pi$ is stochastic, so as to bootstrap the current agent's guess about what actions are good.

To demonstrate the generality of this approach, we use it to enhance the seminal DQN [23] deep RL algorithm. In DQN, the $q$-values are estimated by an online $Q$-network $q_\theta$, with weights copied regularly to a target network $q_{\bar{\theta}}$. The agent behaves following a policy $\pi_\theta \in \mathcal{G}(q_\theta)$ (with $\varepsilon$-greedy exploration), and stores transitions $(s_t, a_t, r_t, s_{t+1})$ in a FIFO replay buffer $\mathcal{B}$. DQN performs

stochastic gradient descent on the loss $\hat{\mathbb{E}}_{\mathcal{B}}[(q_\theta(s_t, a_t) - \hat{q}_{\text{dqn}}(r_t, s_{t+1}))^2]$, regressing the target $\hat{q}_{\text{dqn}}$:

$$\hat{q}_{\text{dqn}}(r_t, s_{t+1}) = r_t + \gamma \sum_{a' \in \mathcal{A}} \pi_{\bar{\theta}}(a'|s_{t+1}) q_{\bar{\theta}}(s_{t+1}, a') \text{ with } \pi_{\bar{\theta}} \in \mathcal{G}(q_{\bar{\theta}}).$$

To derive Munchausen-DQN (M-DQN), we simply modify the regression target. M-RL assumes stochastic policies while DQN computes deterministic policies. A simple way to address this is to not only maximize the return, but also the entropy of the resulting policy, that is adopting the viewpoint of maximum entropy RL [36, 17]. It is straightforward to extend DQN to this setting, see Appx. A.1 for a detailed derivation. We call the resulting agent Soft-DQN (S-DQN). Let $\tau$ be the temperature parameter scaling the entropy, it just amounts to replace the original regression target by

$$\hat{q}_{\text{s-dqn}}(r_t, s_{t+1}) = r_t + \gamma \sum_{a' \in \mathcal{A}} \pi_{\bar{\theta}}(a'|s_{t+1}) \Big( q_{\bar{\theta}}(s_{t+1}, a') - \tau \ln \pi_{\bar{\theta}}(a'|s_{t+1}) \Big) \text{ with } \pi_{\bar{\theta}} = \text{sm}(\tfrac{q_{\bar{\theta}}}{\tau}), \quad (1)$$

where we highlighted the differences with DQN in blue. Notice that this is nothing more than the most straightforward discrete-actions version of Soft Actor-Critic (SAC) [17]. Notice also that in the limit $\tau \to 0$ we retrieve DQN. The last step to obtain M-DQN is to add the scaled log-policy to the reward. Let $\alpha \in [0, 1]$ be a scaling factor, the regression target of M-DQN is thus

$$\hat{q}_{\text{m-dqn}}(r_t, s_{t+1}) = r_t + \alpha \tau \ln \pi_{\bar{\theta}}(a_t|s_t) + \gamma \sum_{a' \in \mathcal{A}} \pi_{\bar{\theta}}(a'|s_{t+1}) \Big( q_{\bar{\theta}}(s_{t+1}, a') - \tau \ln \pi_{\bar{\theta}}(a'|s_{t+1}) \Big), \quad (2)$$

still with $\pi_{\bar{\theta}} = \text{sm}(\tfrac{q_{\bar{\theta}}}{\tau})$, where we highlighted the difference with Soft-DQN in red (retrieved by setting $\alpha = 0$). Hence, M-DQN is genuinely obtained by replacing $\hat{q}_{\text{dqn}}$ by $\hat{q}_{\text{m-dqn}}$ as the regression target of DQN. All details of the resulting algorithm are provide in Appx. B.1.

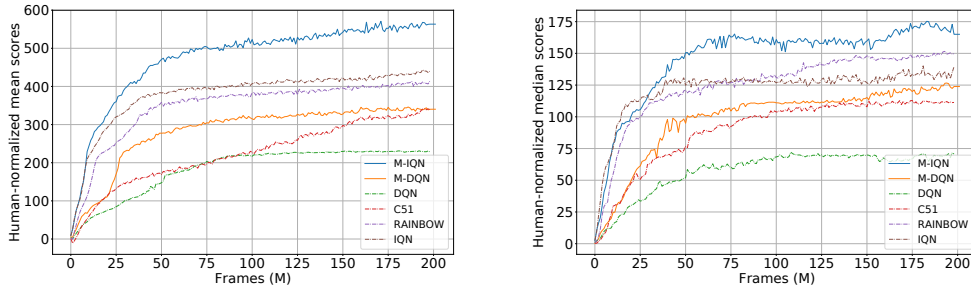

Figure 1: **Left:** Human-normalized mean scores. **Right:** Human-normalized median scores.

Despite being an extremely simple modification of DQN, M-DQN is very efficient. We show in Fig. 10 the Human-normalized mean and median scores for various agents on the full set of 60 Atari games of ALE (more details in Sec. 4). We observe that M-DQN significantly outperforms DQN, but also C51 [8]. As far we know, M-DQN is the first method that is not based on distRL which overtakes C51. These are quite encouraging empirical results.

To demonstrate the versatility of the M-RL principle, we also combine it with IQN [11], a recent and efficient distRL agent (note that IQN has had recent successors, such as Fully Parameterized Quantile Function (FQF) [35], to which in principle, we could also apply M-RL). We denote the resulting algorithm M-IQN. In a nutshell, IQN does not estimate the $q$-function, but the distribution of which the $q$-function is the mean, using a distributional Bellman operator. The (implicit) policy is still greedy according to the $q$-function, computed as the (empirical) mean of the estimated distribution. We apply the exact same recipe: derive soft-IQN using the principle of maximum entropy RL (which is as easy as for DQN), and add the scaled log-policy to the reward. For the sake of showing the generality of our method, we combine M-RL with a version of IQN that uses 3-steps returns (and we compare to IQN and Rainbow, that both use the same). We can observe on Fig. 10 that M-IQN outperforms Rainbow, both in terms of mean and median scores, and thus defines the new state of the art. In addition, even when using only 1-step returns, M-IQN still outperforms Rainbow. This result and the details of M-IQN can be found respectively in Appx. B.3 and B.1.

# 3 What happens under the hood?

The impressive empirical results of M-RL (see Sec. 4 for more) call for some theoretical insights. To provide them, we frame M-DQN in an abstract Approximate Dynamic Programming (ADP) framework and analyze it. We mainly provide two strong results: (1) M-DQN implicitly performs KL regularization between successive policies, which translates in an averaging effect of approximation errors (instead of accumulation in general ADP frameworks); (2) it increases the action-gap by a quantifiable amount which also helps dealing with approximation errors. We also use this section to draw connections with the existing literature in ADP. Let's first introduce some additional notations.

We write $\Delta_X$ the simplex over the finite set $X$ and $Y^X$ the set of applications from $X$ to the set $Y$. With this, an MDP is $\{\mathcal{S}, \mathcal{A}, P \in \Delta_{\mathcal{S}}^{\mathcal{S} \times \mathcal{A}}, r \in \mathbb{R}^{\mathcal{S} \times \mathcal{A}}, \gamma \in (0,1)\}$, the state and action spaces being assumed finite. For $f, g \in \mathbb{R}^{\mathcal{S} \times \mathcal{A}}$, we define a component-wise dot product $\langle f, g \rangle = (\sum_a f(s,a)g(s,a))_s \in \mathbb{R}^{\mathcal{S}}$. This will be used with $q$-functions and (log-) policies, *e.g.* for expectations: $\mathbb{E}_{a \sim \pi(\cdot|s)}[q(s,a)] = \langle \pi, q \rangle(s)$. For $v \in \mathbb{R}^{\mathcal{S}}$, we have $Pv = (\mathbb{E}_{s'|s,a}[v(s')])_{s,a} = (\sum_{s'} P(s'|s,a)v(s'))_{s,a} \in \mathbb{R}^{\mathcal{S} \times \mathcal{A}}$. We also defined a policy-induced transition kernel $P_\pi$ as $P_\pi q = P\langle \pi, q \rangle$. With these notations, the Bellman evaluation operator is $T_\pi q = r + \gamma P_\pi q$ and its unique fixed point is $q_\pi$. An optimal policy still satisfies $\pi_* \in \mathrm{argmax}_{\pi \in \Delta_{\mathcal{A}}^{\mathcal{S}}} q_\pi$. The set of greedy policies can be written as $\mathcal{G}(q) = \mathrm{argmax}_{\pi \in \Delta_{\mathcal{A}}^{\mathcal{S}}} \langle \pi, q \rangle$. We'll also make use of the entropy of a policy, $\mathcal{H}(\pi) = -\langle \pi, \ln \pi \rangle$, and of the KL between two policies, $\mathrm{KL}(\pi_1 || \pi_2) = \langle \pi_1, \ln \pi_1 - \ln \pi_2 \rangle$.

A softmax is the maximizer of the Legendre-Fenchel transform of the entropy [9, 32], $\mathrm{sm}(q) = \mathrm{argmax}_\pi \langle \pi, q \rangle + \mathcal{H}(\pi)$. Using this and the introduced notations, we can write M-DQN in the following abstract form (each iteration consists of a greedy step and an evaluation step):

$$\begin{cases} \pi_{k+1} = \mathrm{argmax}_{\pi \in \Delta_{\mathcal{A}}^{\mathcal{S}}} \langle \pi, q_k \rangle + \tau \mathcal{H}(\pi) \\ q_{k+1} = r + \alpha\tau \ln \pi_{k+1} + \gamma P \langle \pi_{k+1}, q_k - \tau \ln \pi_{k+1} \rangle + \epsilon_{k+1}. \end{cases} \qquad \text{M-VI}(\alpha, \tau) \qquad (3)$$

We call the resulting scheme Munchausen Value Iteration, or M-VI($\alpha, \tau$). The term $\epsilon_{k+1}$ stands for the error between the actual and the ideal update (sampling instead of expectation, approximation of $q_k$ by a neural network, fitting of the neural network). Removing the red term, we retrieve approximate VI (AVI) regularized by a scaled entropy, as introduced by Geist et al. [15], of which Soft-DQN is an instantiation (as well as SAC, with additional error in the greedy step). Removing also the blue term, we retrieve the classic AVI [26], of which DQN is an instantiation.

To get some insights, we rewrite the evaluation step, setting $\alpha = 1$ and with $q'_k \triangleq q_k - \tau \ln \pi_k$:

$$q_{k+1} = r + \tau \ln \pi_{k+1} + \gamma P \langle \pi_{k+1}, q_k - \tau \ln \pi_{k+1} \rangle + \epsilon_{k+1}$$

$$\Leftrightarrow q_{k+1} - \tau \ln \pi_{k+1} = r + \gamma P \langle \pi_{k+1}, q_k - \tau \ln \pi_k - \tau \ln \frac{\pi_{k+1}}{\pi_k} \rangle + \epsilon_{k+1}$$

$$\Leftrightarrow q'_{k+1} = r + \gamma P (\langle \pi_{k+1}, q'_k \rangle - \tau \mathrm{KL}(\pi_{k+1} || \pi_k)) + \epsilon_{k+1}.$$

Then, the greedy step can be rewritten as (looking at what $\pi_{k+1}$ maximizes)

$$\langle \pi, q_k \rangle + \tau \mathcal{H}(\pi) = \langle \pi, q'_k + \tau \ln \pi_k \rangle - \tau \langle \pi, \ln \pi \rangle = \langle \pi, q'_k \rangle - \tau \mathrm{KL}(\pi || \pi_k). \qquad (4)$$

We have just shown that M-VI($1, \tau$) implicitly performs KL regularization between successive policies.

This is a very insightful result as KL regularization is the core component of recent efficient RL agents such as TRPO [27] or MPO [2]. It is extensively discussed by Vieillard et al. [32]. Interestingly, we can show that the sequence of policies produced by M-VI($\alpha, \tau$) is the same as the one of their Mirror Descent VI (MD-VI), with KL scaled by $\alpha\tau$ and entropy scaled by $(1-\alpha)\tau$. Thus, M-VI($\alpha, \tau$) is equivalent to MD-VI($\alpha\tau, (1-\alpha)\tau$), as formalized below (proof in Appx. A.2).

**Theorem 1.** *For any $k \geq 0$, define $q'_k = q_k - \alpha\tau \ln \pi_k$, we have*

$$(3) \Leftrightarrow \begin{cases} \pi_{k+1} = \mathrm{argmax}_{\pi \in \Delta_{\mathcal{A}}^{\mathcal{S}}} \langle \pi, q'_k \rangle - \alpha\tau \mathrm{KL}(\pi || \pi_k) + (1-\alpha)\tau \mathcal{H}(\pi) \\ q'_{k+1} = r + \gamma P (\langle \pi_{k+1}, q'_k \rangle - \alpha\tau \mathrm{KL}(\pi_{k+1} || \pi_k) + (1-\alpha)\tau \mathcal{H}(\pi_{k+1})) + \epsilon_{k+1} \end{cases}.$$

*Moreover, [32, Thm. 1] applies to M-VI(1,$\tau$) and [32, Thm. 2] applies to M-VI($\alpha < 1, \tau$).*

In their work, Vieillard et al. [32] show that using regularization can reduce the dependency to the horizon $(1-\gamma)^{-1}$ and that using a KL divergence allows for a compensation of the errors $\epsilon_k$ over

iterations, which is not true for classical ADP. We refer to them for a detailed discussion on this topic. However, we would like to highlight that they acknowledge that *their theoretical analysis does not apply to the deep RL setting*. The reason being that their analysis does not hold when the greedy step is approximated, and they deem as impossible to do the greedy step exactly when using neural network. Indeed, computing $\pi_{k+1}$ by maximizing eq. (4) yields an analytical solution proportional to $\pi_k \exp(\frac{q_k}{\tau})$, and that thus depends on the previous policy $\pi_k$. Consequently, the solution to this equation cannot be computed exactly when using deep function approximation (unless one would be willing to remember every computed policy). On the contrary, *their analysis applies in our deep RL setting*. In M-VI, the KL regularization is implicit, so *we do not introduce errors in the greedy step*. To be precise, the greedy step of M-VI is only a softmax of the $q$-function, which can be computed exactly in a discrete actions setting, even when using deep networks. Their strong bounds for MD-VI therefore hold for M-VI, as formalized in Thm. 1, and in particular for M-DQN.

Indeed, let $q_{\bar{\theta}_k}$ be the $k^{\text{th}}$ update of the target network, write $q_k = q_{\bar{\theta}_k}$, $\pi_{k+1} = \text{sm}(\frac{q_k}{\tau})$, and define $\epsilon_{k+1} = q_{k+1} - (r + \alpha \ln \pi_{k+1} - \gamma P \langle \pi_{k+1}, q_k - \tau \ln \pi_{k+1} \rangle)$, the difference between the actual update and the ideal one. As a direct corollary of Thm. 1 and [32, Thm. 1], we have that, for $\alpha = 1$,

$$\|q_* - q_{\pi_k}\|_\infty \leq \frac{2}{1-\gamma} \left\| \frac{1}{k} \sum_{j=1}^{k} \epsilon_j \right\|_\infty + \frac{4}{(1-\gamma)^2} \frac{r_{\max} + \tau \ln |\mathcal{A}|}{k},$$

with $r_{\max}$ the maximum reward (in absolute value), and with $q_{\pi_k}$ the true value function of the policy of M-DQN. This is a very strong bound. The error term is $\|\frac{1}{k} \sum_{j=1}^{k} \epsilon_j\|_\infty$, to be compared to the one of AVI [26], $(1-\gamma) \sum_{j=1}^{k} \gamma^{k-j} \|\epsilon_j\|_\infty$. Instead of having a discounted sum of the norms of the errors, we have the norm of the average of the errors. This is very interesting, as it allows for a compensation of errors between iterations instead of an accumulation (sum and norm do not commute). The error term is scaled by $(1-\gamma)^{-1}$ (the average horizon of the MDP), while the one of AVI would be scaled by $(1-\gamma)^{-2}$. This is also quite interesting, a $\gamma$ close to 1 impacts less negatively the bound. We refer to [32, Sec. 4.1] for further discussions about the advantage of this kind of bounds. Similarly, we could derive a bound for the case $\alpha < 1$, and even more general and meaningful component-wise bounds. We defer the statement of these bounds and their proofs to Appx. A.3.

From Eq. (3), we can also relate the proposed approach to another part of the literature. Still from basic properties of the Legendre-Fenchel transform, we have that $\max_\pi \langle q, \pi \rangle + \tau \mathcal{H}(\pi) = \langle \pi_{k+1}, q_k \rangle + \tau \mathcal{H}(\pi_{k+1}) = \ln \langle 1, \exp q \rangle$. In other words, if the maximizer is the softmax, the maximum is the log-sum-exp. Using this, Eq. (3) can be rewritten as (see Appx. A.4 for a detailed derivation)

$$q_{k+1} = r + \gamma P(\tau \ln \langle 1, \exp \frac{q_k}{\tau} \rangle) + \alpha(q_k - \tau \ln \langle 1, \exp \frac{q_k}{\tau} \rangle) + \epsilon_{k+1}. \tag{5}$$

This is very close to Conservative Value Iteration[2] (CVI) [20], a purely theoretical algorithm, as far as we know. With $\alpha = 0$ (without Munchausen), we get Soft Q-learning [14, 16]. Notice that with this, CVI can be seen as soft $Q$-learning plus a scaled and smooth advantage (the term $\alpha(q_k - \tau \ln \langle 1, \exp \frac{q_k}{\tau} \rangle)$). With $\alpha = 1$, we retrieve a variation of Dynamic Policy Programming (DPP) [3, Appx. A]. DPP has been extended to a deep learning setting [30], but it is less efficient than DQN[3] [32]. Taking the limit $\tau \to 0$, we retrieve Advantage Learning (AL) [5, 7] (see Appx. A.4):

$$q_{k+1} = r + \gamma P \langle \pi_{k+1}, q_k \rangle + \alpha(q_k - \langle \pi_{k+1}, q_k \rangle) + \epsilon_{k+1} \text{ with } \pi_{k+1} \in \mathcal{G}(q_k). \tag{6}$$

AL aims at increasing the action-gap [13] defined as the difference, for a given state, between the (optimal) value of the optimal action and that of the suboptimal ones. The intuitive reason to want a large action-gap is that it can mitigate the undesirable effects of approximation and estimation errors made on $q$ on the induced greedy policies. Bellemare et al. [7] have introduced a family of Bellman-like operators that are gap-increasing. Not only we show that M-VI is gap-increasing but we also quantify the increase. To do so, we introduce some last notations. As we explained before, with $\alpha = 0$, M-VI$(0, \tau)$ reduces to AVI regularized by an entropy (that is, maximum entropy RL). Without error, it is known that the resulting regularized MDP has a unique optimal policy $\pi_*^\tau$ and a unique optimal $q$-function[4] $q_*^\tau$ [15]. This being defined, we can state our result (proven in Appx. A.5).

**Theorem 2.** *For any state $s \in \mathcal{S}$, define the action-gap of an MPD regularized by an entropy scaled by $\tau$ as $\delta_*^\tau(s) = \max_a q_*^\tau(s,a) - q_*^\tau(s,\cdot) \in \mathbb{R}_+^\mathcal{A}$. Define also $\delta_k^{\alpha,\tau}(s)$ as the action-gap for the $k^{th}$ iteration of M-VI($\alpha,\tau$), without error ($\epsilon_k = 0$): $\delta_k^{\alpha,\tau}(s) = \max_a q_k(s,a) - q_k(s,\cdot) \in \mathbb{R}_+^\mathcal{A}$. Then, for any $s \in \mathcal{S}$, for any $0 \leq \alpha \leq 1$ and for any $\tau > 0$, we have*

$$\lim_{k \to \infty} \delta_k^{\alpha,\tau}(s) = \frac{1+\alpha}{1-\alpha} \delta_*^{(1-\alpha)\tau}(s),$$

*with the convention that $\infty \cdot 0 = 0$ for $\alpha = 1$.*

Thus, the original action-gap is multiplied by $\frac{1+\alpha}{1-\alpha}$ with M-VI. In the limit $\alpha = 1$, it is even infinite (and zero for the optimal actions). This suggests choosing a large value of $\alpha$, but not too close to 1 (for numerical stability: if having a large action-gap is desirable, having an infinite one is not).

## 4 Experiments

**Munchausen agents.** We implement M-DQN and M-IQN as variations of respectively DQN and IQN from Dopamine [10]. We use the same hyperparameters for IQN[5], and we only change the optimizer from RMSProp to Adam for DQN. This is actually not anodyne, and we study its impact in an ablation study. We also consider a Munchausen-specific modification, *log-policy clipping*. Indeed, the log-policy term is not bounded, and can cause numerical issues if the policy becomes too close to deterministic. Thus, with a hyperparameter $l_0 < 0$, we replace $\tau \ln \pi(a|s)$ by $[\tau \ln \pi(a|s)]_{l_0}^0$, where $[\cdot]_x^y$ is the clipping function. For numerical stability, we use a specific log-sum-exp trick to compute the log-policy (see App. B.1). Hence, we add three parameters to the modified agent: $\alpha, \tau$ and $l_0$. After some tuning on a few Atari games, we found a working zone for these parameters to be $\alpha = 0.9$, $\tau = 0.03$ and $l_0 = -1$, used for all experiments, in M-DQN and M-IQN. All details about the rest of the parameters can be found in Appx. B.1. DQN and IQN use $\varepsilon$-greedy policies to interact with the environment. Although M-DQN and M-IQN produce naturally stochastic policies, we use the same $\varepsilon$-greedy policies. We discuss this further in Appx. B.2, where we also compare to stochastic policies.

**Baselines.** First, we consider both DQN and IQN, as these are the algorithms we modify. Second, we compare to C51 because, as far as we know, it has never been outperformed by a non-distRL agent before. We also consider Rainbow, as it stands for being the state-of-the-art non-distributed agent on ALE. All our baselines are taken from Dopamine. For Rainbow, this version doesn't contain all the original improvements, but only the ones deemed as the more important and efficient by Hessel et al. [18]: $n$-steps returns and Prioritized Experience Replay (PER) [25], on top of C51.

**Task.** We evaluate our methods and the baselines in the ALE environment, *i.e.* on the full set of 60 Atari games. Notice that it is not a "canonical" environment. For example, choosing to end an episode when an agent loses a life or after game-over can dramatically change the score an agent can reach (*e.g.*, [10, Fig. 4]). The same holds for using sticky actions, introducing stochasticity in the dynamics (*e.g.*, [10, Fig. 6]). Even the ROMs could be different, with unpredictable consequences (*e.g.* different video encoding). Here, we follow the methodological best practices proposed by Machado et al. [22] and instantiated in Dopamine [10], that also makes the ALE more challenging. Notably, the results we present are hardly comparable to the ones presented in the original publications of DQN [23], C51 [8], Rainbow [18] or IQN [11], that use a different, easier, setting. Yet, for completeness, we report results on one game (Asterix) using an ALE setting as close as possible to the original papers, in Appx. B.4: the baseline results match the previously published ones, and M-RL still raises improvement. We also highlight that we stick to a single-agent version of the environment: we do not claim that our method can be compared to highly distributed agents, such as R2D2 [19] or Agent57 [4], that use several versions of the environment in parallel, and train on a much higher number of frames (around 10G frames vs 200M here). Yet, we are confident that our approach could easily apply to such agents.

**Metrics.** All algorithms are evaluated on the same training regime (details in Appx.B.1), during 200M frames, and results are averaged over 3 seeds. As a metric for any games, we compute the "baseline-normalized" score, for each iteration (here, 1M frames), normalized so that $0\%$ corresponds to a random score, and $100\%$ to the final performance of the baseline. At each iteration, the score is

the undiscounted sum of rewards, averaged over the last 100 learning episodes. The normalized score is then $\frac{a-r}{|b-r|}$, with $a$ the score of the agent, $b$ the score of the baseline, and $r$ the score of a random policy. For a human baseline, the scores are those provided in Table 3 (Appx. B.6), for an agent baseline the score is the one after 200M frames. With this, we provide aggregated results, showing the mean and the median over games, as learning proceeds when the baseline is the human score (*e.g.*, Fig. 1), or after 200M steps with human and Rainbow baselines in Tab. 3 (more results in Appx. B.6, as learning proceeds). We also compute a "baseline-improvement" score as $\frac{a-b}{|b-r|}$, and use it to report a per-game improvement after 200M frames (Fig. 4, M-Agent versus Agent, or Appx. B.6).

**Action-gap.** We start by illustrating the action-gap phenomenon suggested by Thm. 2. To do so, let $q_\theta$ be the $q$-function of a given agent after training for 200M steps. At any time-step $t$, write $\hat{a}_t \in \mathrm{argmax}_{a \in \mathcal{A}}\, q_\theta(s_t, a)$ the current greedy action, we compute the empirical action-gap as the difference of estimated values between the best and second best actions, $q_\theta(s_t, \hat{a}_t) - \max_{a \in \mathcal{A} \setminus \{\hat{a}_t\}} q_\theta(s_t, a)$. We do so for M-DQN, for AL (that was introduced specifically to increase the action-gap) and for DQN with Adam optimizer (Adam DQN), as both build on top of it (only changing the regression targets, see Appx. B.1 for details).

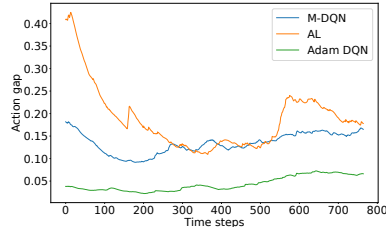

Figure 2: Action-gaps (Asterix).

We consider the game Asterix, for which the final average performance of the agents are (roughly) 15k for Adam DQN, 13k for AL and 20k for M-DQN. We report the results on Fig. 2: we run each agent for 10 trajectories, and average the resulting action-gaps (the length of the resulting trajectory is the one of the shorter trajectory, we also apply an exponential smoothing of 0.99). Both M-DQN and AL increase the action-gaps compared to Adam DQN. If AL increases it more, it seems also to be less stable, and less proportional to the original action-gap. Despite this increase, it performs worse than Adam DQN (13k vs 15k), while M-DQN increases it and performs better (20k vs 15k). An explanation to this phenomenon could the one of Van Seijen et al. [31], who suggest that what is important is not the value of the action gap itself, but its uniformity over the state-action space: here, M-DQN seems to benefit from a more stable action-gap than AL. This figure is for an illustrative purpose, one game is not enough to draw conclusions. Yet, the following ablation shows that globally M-DQN performs better than AL. Also, it benefits from more theoretical justifications (not only quantified action-gap increase, but also implicit KL-regularization and resulting performance bounds).

**Ablation study.** We've build M-DQN from DQN by adding the Adam optimizer (Adam DQN), extending it to maximum entropy RL (Soft-DQN, Eq. (1)), and then adding the Munchausen term (M-DQN, Eq. (2)). A natural ablation is to remove the Munchausen term, and use only maximum entropy RL, by considering M-DQN with $\alpha = 0$ (instead of 0.9 for M-DQN), and the same $\tau$ (here, $3e-2$), which would give Soft-DQN($\tau$). However, Thm. 1 states that M-DQN performs entropy regularization with an implicit coefficient of $(1-\alpha)\tau$, so to compare M-DQN and Soft-DQN fairly, one should evaluate Soft-DQN with such a temperature, that is $3e-3$ in this case. We denote this ablation as Soft-DQN($(1-\alpha)\tau$). As sketched in Sec. 3, AL can also be seen as a limit case (on an abstract way, as $\tau \to 0$, see also Appx. B.1 for details on the algorithm). We provide an ablation study of all these variations, all using Adam (except DQN), in Fig. 3. All methods perform better than DQN. Adam DQN performs very well and is even competitive with C51. This is an interesting insight, as changing the optimizer compared to the published parameters dramatically improves the performance, and Adam DQN could be considered as a better baseline[6]. Surprisingly, if better than DQN, Soft-DQN does not perform better than Adam DQN. This suggests that maximum entropy RL alone might not be sufficient. We kept the temperature $\tau = 0.03$, and one could argue that it was not tuned for Soft DQN, but it is on par with the temperature of similar algorithms [28, 32]. We observe that AL performs better than Adam DQN. Again, we kept $\alpha = 0.9$, but this is consistent with the best performing parameter of Bellemare et al. [7, *e.g.*, Fig. 7]. The proposed M-DQN outperforms all other methods, both in mean and median, and especially Soft-DQN by a significant margin (the sole difference being the Munchausen term).

**Comparison to the baselines.** We report aggregated results as Human-normalized mean and median scores on Figure 1, that compares the Munchausen agents to the baselines. M-DQN is largely

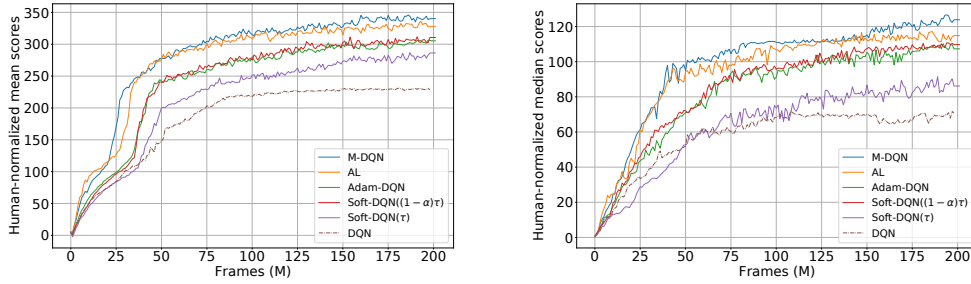

Figure 3: Ablation study of M-DQN: Human-normalized mean (**left**) and median (**right**) scores.

Table 1: Mean/median Human/Rainbow-normalized scores at 200M frames, on the 60 games, averaged over 3 random seeds. In **bold** are the best of each column, and in blue over Rainbow. We also provide the number of improved games (compared to Human and Rainbow).

|  | Human-normalized | | | Rainbow-normalized | | |
|---|---|---|---|---|---|---|
|  | Mean | Median | #Improved | Mean | Median | #Improved |
| M-DQN | 340% | 124% | 37 | 89% | 92% | 21 |
| M-IQN | **563%** | **165%** | **43** | **130%** | **109%** | **38** |
| RAINBOW | 414% | 150% | 43 | 100% | 100% | - |
| IQN | 441% | 139% | 41 | 105% | 99% | 27 |
| C51 | 339% | 111% | 33 | 84% | 70% | 11 |
| DQN | 228% | 71% | 23 | 51% | 51% | 3 |

over DQN, and outperforms C51 both in mean and median. It is remarkable that M-DQN, justified by theoretically sound RL principles and without using common deep RL tricks like $n$-steps returns, PER or distRL, is competitive with distRL methods. It is even close to IQN (in median), considered as the best distRL-based agent. We observe that M-IQN, that combines IQN with Munchausen principle, is better than all other baselines, by a significant margin in mean. We also report the final Human-normalized and Rainbow-normalized scores of all the algorithms in Table 1. These results are on par with the Human-normalized scores of Fig. 1 (see Appx. B.6 for results over frames). M-DQN is still close to IQN i median, is better than DQN, and C51, while M-IQN is the best agent w.r.t. all metrics.

**Per-game improvements.** In Figure 4, we report the improvement for each game of the Munchausen agents over the algorithms they modify. The "Munchausened" versions show significant improvements, on a large majority of Atari games (53/60 for M-DQN vs DQN, 40/60 for M-IQN vs IQN). This result also explains the sometime large difference between the mean and median metrics, as some games benefit from a particularly large improvement. All learning curves are in Appx B.6.

## 5   Conclusion

In this work, we presented a simple extension to RL algorithms: Munchausen RL. This method augments the immediate rewards by the scaled logarithm of the policy computed by an RL agent. We applied this method to a simple variation of DQN, Soft-DQN, resulting in the M-DQN algorithm. M-DQN shows large performance improvements: it outperforms DQN on 53 of the 60 Atari games, while simply using a modification of the DQN loss. In addition, it outperforms the seminal distributional RL algorithm C51. We also extended the Munchausen idea to distributional RL, showing that it could be successfully combined with IQN to outperform the Rainbow baseline. Munchausen-DQN relies on theoretical foundations. To show that, we have studied an abstract Munchausen Value Iteration scheme and shown that it implicitly performs KL regularization. Notably, the strong theoretical results of [32] apply to M-DQN. By rewriting it in an equivalent ADP form, we have related our approach to the literature, notably to CVI, DPP and AL . We have shown that M-VI increases the

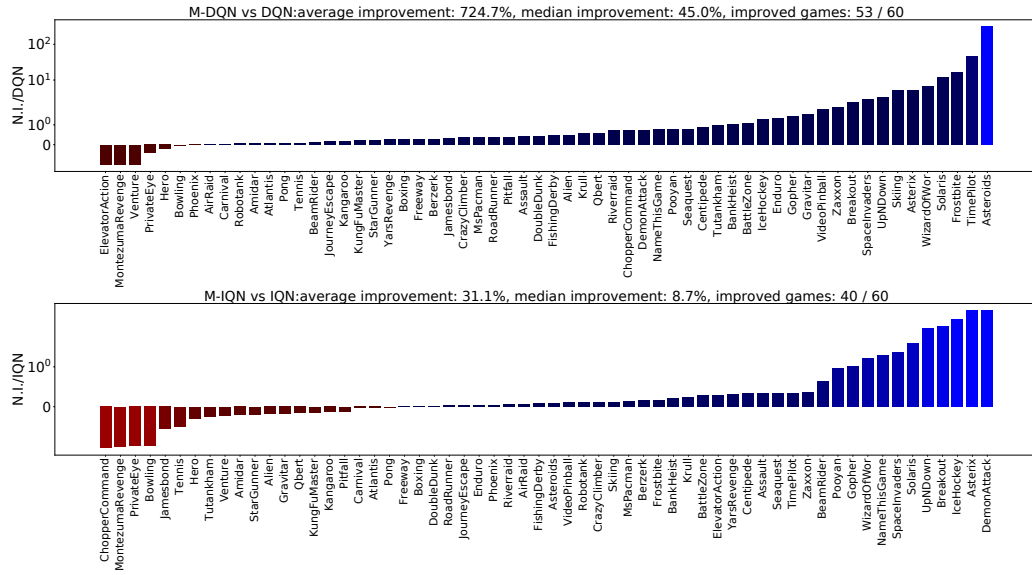

Figure 4: Per-game improvement of M-DQN vs DQN (**top**) and of M-IQN vs IQN (**bottom**).

action-gap, and we have quantified this increase, that can be infinite in the limit. In the end, this work highlights that a thoughtful revisiting of the core components of reinforcement learning can lead to new and efficient deep RL algorithms.

## Broader impact

The core contribution of this work is to propose a new RL algorithm, that surpasses state of the art results on a challenging discrete actions environment. We believe it can impact positively the RL community, as it shades light on fundamental ideas, justified by deep theoretical foundations, that proves to be efficient in practice. Outside of the RL community, the impact of this paper is part of the global impact of RL. This work is mainly algorithmic and theoretical, with no specific applications in mind, but participates to the general development of efficient and practical RL methods.

## Funding transparency statement

Nothing to disclose.

## Footnotes

[1]It appears that the benefits of distRL do not really come from RL principles, but rather from the regularizing effect of modelling a distribution and its role as an auxiliary task in a deep learning context [21].

[2]In CVI, $\langle 1, \exp \frac{q_k}{\tau} \rangle$ is replaced by $\langle \frac{1}{|\mathcal{A}|}, \exp \frac{q_k}{\tau} \rangle$.

[3]In fact, Tsurumine et al. [30] show better performance for deep DPP than for DQN in their setting. Yet, their experiment involves a small number of interactions, while the function estimated by DPP is naturally diverging. See [33, Sec. 6] for further discussion about this.

[4]It can be related to the unregularized optimal $q$-function, $\|q_*^\tau - q_*\|_\infty \leq \frac{\tau \ln |\mathcal{A}|}{1-\gamma}$ [15].

[5]By default, Dopamine's IQN uses 3-steps returns. We rather consider 1-step returns, as in [11].

[6]To be on par with the literature, we keep using the published DQN as the baseline for other experiments.

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
