[Supplementary Material]

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

[7]The setting is still not exactly the same, due to less enhancements in the Dopamine's Rainbow, a different codebase, but also a difference in the start (human start vs no-op for Rainbow, straight start for us), and possibly a different ROM, which cannot be checked.

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

# A  Detailed derivation and proofs

This appendix provides additional details regarding the derivation sketched in the main paper as well as the proofs of the stated results:

- Appx. A.1 details the derivation of Soft-DQN.

- Appx. A.2 proves the result that relates Munchausen VI to Mirror Descent VI.

- Appx. A.3 provides and proves component-wise bounds for Munchausen VI, that also apply to Munchausen-DQN.

- Appx. A.4 details the derivation that allows linking the proposed Munchausen approach to the literature.

- Appx. A.5 proves the result quantifying the increase of the action-gap.

First, we recall the notations introduced in the main paper as well as some useful facts about (regularized) MDPs.

We write $\Delta_X$ the simplex over the finite set $X$ and $Y^X$ the set of applications from $X$ to the set $Y$. An MDP is a tuple $\{\mathcal{S}, \mathcal{A}, P, r, \gamma\}$, with $\mathcal{S}$ and $\mathcal{A}$ the state and action spaces (here assumed finite), $P \in \Delta_{\mathcal{S}}^{\mathcal{S} \times \mathcal{A}}$ the Markovian transition kernel, $r \in \mathbb{R}^{\mathcal{S} \times \mathcal{A}}$ the reward function, uniformly bounded by $r_{\max}$, and $\gamma \in (0, 1)$ the discount factor. A policy $\pi \in \Delta_{\mathcal{A}}^{\mathcal{S}}$ associates to each state a distribution over actions (a deterministic policy being a special case), and the quality of a policy is quantified by the expected discounted cumulative return, formalized as the state-action value function, $q_\pi(s, a) = \mathbb{E}_\pi[\sum_{t=0}^{\infty} \gamma^t r(s_t, a_t) | s_0 = s, a_0 = a]$, the expectation being over trajectories induced by the policy $\pi$ and the dynamics.

For $f, g \in \mathbb{R}^{\mathcal{S} \times \mathcal{A}}$, we define a component-wise dot product $\langle f, g \rangle = (\sum_a f(s, a) g(s, a))_s \in \mathbb{R}^{\mathcal{S}}$. This will be used with $q$-functions and (log-) policies. For $v \in \mathbb{R}^{\mathcal{S}}$, we have $Pv = (\mathbb{E}_{s'|s,a}[v(s')])_{s,a} \in \mathbb{R}^{\mathcal{S} \times \mathcal{A}}$. We also defined a policy-induced transition kernel $P_\pi$ as $P_\pi q = P\langle \pi, q \rangle$. With this, the Bellman evaluation operator is $T_\pi q = r + \gamma P_\pi q$ and its unique fixed point is $q_\pi$.

An optimal policy satisfies $\pi_* \in \mathrm{argmax}_\pi q_\pi$, component-wise, and the associated (unique) optimal value function $q_* = q_{\pi_*}$ satisfies the Bellman equation $q_*(s, a) = r(s, a) + \gamma E_{s'|s,a}[\max_{a'} q_*(s', a')]$. We write the set of greedy policies as $\mathcal{G}(q) = \mathrm{argmax}_{\pi \in \Delta_{\mathcal{A}}^{\mathcal{S}}} \langle \pi, q \rangle$. We'll also use softmax policies, $\pi = \mathrm{sm}(q) \Leftrightarrow \pi(a|s) = \frac{\exp q(s,a)}{\sum_{a'} \exp q(s,a')}$.

We'll also make use of the entropy of a policy, $\mathcal{H}(\pi) = -\langle \pi, \ln \pi \rangle$, and of the KL between two policies, $\mathrm{KL}(\pi_1 \| \pi_2) = \langle \pi_1, \ln \pi_1 - \ln \pi_2 \rangle$. An MDP regularized by a scaled entropy $\tau \mathcal{H}(\pi)$, also known as maximum entropy RL, optimizes for the reward $r - \tau \ln \pi$. It has a unique optimal $q$-function $q_*^\tau$ and a unique optimal policy $\pi_*^\tau$, related by $\pi_*^\tau = \mathrm{sm}(q_*^\tau)$; it is related to the solution of the unregularized MDP by $\|q_*^\tau - q_*\|_\infty \leq \frac{\tau \ln |\mathcal{A}|}{1-\gamma}$ [15]. We also write $q_\pi^\tau$ the value function of the policy $\pi$ in this regularized MDP.

Lastly, by classic properties of the Legendre-Fenchel transform [9, 32], we have $\forall q \in \mathbb{R}^{\mathcal{S} \times \mathcal{A}}$:

$$\max_{\pi \in \Delta_{\mathcal{A}}^{\mathcal{S}}} \langle q, \pi \rangle + \tau \mathcal{H}(\pi) = \tau \ln \langle 1, \exp \frac{q}{\tau} \rangle = \langle \pi', q \rangle + \tau \mathcal{H}(\pi') \text{ with } \pi' = \mathrm{sm}(q).$$

## A.1 Derivation of Soft-DQN

Soft-DQN can be derived from the maximum entropy RL framework. To do so, it is sufficient to follows the derivation that Haarnoja et al. [17] made for SAC. In our case, the actions being discrete, no approximation is necessary for computing the policy (there is no actor), which gives Soft-DQN.

Alternatively, and equivalently, one can derive Soft-DQN as an approximate VI scheme for an MDP regularized by a scaled entropy. The regularized VI scheme is [15, 32]:

$$\begin{cases} \pi_{k+1} = \mathrm{argmax}_\pi \langle \pi, q_k \rangle + \tau \mathcal{H}(\pi) \\ q_{k+1} = r + \gamma P(\langle \pi_{k+1}, q_k \rangle + \tau \mathcal{H}(\pi_{k+1})) + \epsilon_{k+1} \end{cases}.$$

From Legendre-Fenchel, $\pi_{k+1} = \mathrm{sm}(q_k)$. Using basic calculus, we have

$$\langle \pi_{k+1}, q_k \rangle + \tau \mathcal{H}(\pi_{k+1}) = \langle \pi_{k+1}, q_k \rangle - \tau \langle \pi_{k+1}, \ln \pi_{k+1} \rangle = \langle \pi_{k+1}, q_k - \tau \ln \pi_{k+1} \rangle.$$

Thus, we can write equivalently the regularized VI scheme as

$$q_{k+1} = r + \gamma P \langle \pi_{k+1}, q_k - \tau \ln \pi_{k+1} \rangle + \epsilon_{k+1}, \text{ with } \pi_{k+1} = \mathrm{sm}(q_k),$$

which is basically the Soft-DQN target depicted in Eq. (1).

## A.2 Proof of Thm. 1

The proof is similar to the one done in the main paper for the case $\alpha = 1$. Recall Eq. (3), that gives an iteration of M-VI($\alpha,\tau$):

$$\begin{cases} \pi_{k+1} = \mathrm{argmax}_{\pi \in \Delta_{\mathcal{A}}^{\mathcal{S}}} \langle \pi, q_k \rangle + \tau \mathcal{H}(\pi) \\ q_{k+1} = r + \alpha \tau \ln \pi_{k+1} + \gamma P \langle \pi_{k+1}, q_k - \tau \ln \pi_{k+1} \rangle + \epsilon_{k+1}. \end{cases}$$

Define for any $k \geq 0$ the term $q_k'$ as

$$q_k' = q_k - \alpha \tau \ln \pi_k.$$

By basic calculus, we can rewrite the evaluation step as follows:

$$\begin{aligned} q_{k+1} &= r + \alpha \tau \ln \pi_{k+1} + \gamma P \langle \pi_{k+1}, q_k - \tau \ln \pi_{k+1} \rangle + \epsilon_{k+1} \\ &= r + \alpha \tau \ln \pi_{k+1} + \gamma P \langle \pi_{k+1}, q_k - \alpha \tau \ln \pi_k + \alpha \tau \ln \pi_k - \tau \ln \pi_{k+1} \rangle + \epsilon_{k+1} \\ \Leftrightarrow q_{k+1}' &= r + \gamma P \langle \pi_{k+1}, q_k' - \alpha \tau \ln \frac{\pi_{k+1}}{\pi_k} - (1-\alpha)\tau \ln \pi_{k+1} \rangle + \epsilon_{k+1} \\ &= r + \gamma P \left( \langle \pi_{k+1}, q_k' \rangle - \alpha \tau \, \mathrm{KL}(\pi_{k+1} || \pi_k) + (1-\alpha)\tau \mathcal{H}(\pi_{k+1}) \right) + \epsilon_{k+1}. \end{aligned}$$

For the greedy step, we have:

$$\begin{aligned} \langle \pi, q_k \rangle + \tau \mathcal{H}(\pi) &= \langle \pi, q_k - \tau \ln \pi \rangle \\ &= \langle \pi, q_k' + \alpha \tau \ln \pi_k - \tau \ln \pi \rangle \\ &= \langle \pi, q_k' - \alpha \tau \ln \frac{\pi}{\pi_k} - (1-\alpha)\tau \ln \pi \rangle \\ &= \langle \pi, q_k' \rangle - \alpha \tau \, \mathrm{KL}(\pi || \pi_k) + (1-\alpha)\tau \mathcal{H}(\pi). \end{aligned}$$

Therefore, we have shown that

$$\begin{cases} \pi_{k+1} = \mathrm{argmax}_{\pi \in \Delta_{\mathcal{A}}^{\mathcal{S}}} \langle \pi, q_k \rangle + \tau \mathcal{H}(\pi) \\ q_{k+1} = r + \alpha \tau \ln \pi_{k+1} + \gamma P \langle \pi_{k+1}, q_k - \tau \ln \pi_{k+1} \rangle + \epsilon_{k+1} \end{cases}$$

$$\Updownarrow$$

$$\begin{cases} \pi_{k+1} = \mathrm{argmax}_{\pi \in \Delta_{\mathcal{A}}^{\mathcal{S}}} \langle \pi, q_k' \rangle - \alpha \tau \, \mathrm{KL}(\pi || \pi_k) + (1-\alpha)\tau \mathcal{H}(\pi) \\ q_{k+1}' = r + \gamma P \left( \langle \pi_{k+1}, q_k' \rangle - \alpha \tau \, \mathrm{KL}(\pi_{k+1} || \pi_k) - (1-\alpha)\tau \mathcal{H}(\pi_{k+1}) \right) + \epsilon_{k+1} \end{cases}.$$

This is exactly the update rule of MD-VI($\alpha\tau, (1-\alpha)\tau$) by Vieillard et al. [32]. Initialized with the same policy $\pi_0$ and such that $q_0' = q_0 - \tau \ln \pi_0$, both algorithms will produce the same sequence of policies (for the same sequence of errors). This is enough for [32, Thm. 1] to apply to M-VI($1,\tau$), producing the same sequence of policies that MD-VI($\tau,0$), the result bounding component-wise $q_* - q_{\pi_k}$ (it only involves the computed policy). This is also enough for [32, Thm. 2] to apply to M-VI($\alpha,\tau$), producing the same sequence of policies that MD-VI($\alpha\tau, (1-\alpha)\tau$), the result bounding component-wise $q_*^{(1-\alpha)\tau} - q_{\pi_k}$.

### A.3 Component-wise bounds for Munchausen VI

We state the component-wise bounds for M-VI, announced in Sec. 3. We recall that they apply to M-DQN, as explained in Sec. 3 (by defining to what corresponds $q_k$ and $\epsilon_k$ for M-DQN). First, we provide a bound for the case $\alpha = 1$.

**Corollary 1.** *Let $(q_k, \pi_k)_{k \geq 0}$ be the sequence of q-functions and policies produced by M-VI(1,$\tau$), with $\pi_0$ the uniform policy and $q_0$ such that $\|q_0 - \tau \ln \pi_0\|_\infty \leq \frac{r_{max}}{1-\gamma}$. Define*

$$E_k = -\sum_{j=1}^{k} \epsilon_j,$$

$$\text{and } A_k^1 = (I - \gamma P_{\pi_*})^{-1} - (I - \gamma P_{\pi_k})^{-1}.$$

*Assume that $\|q_k - \tau \ln \pi_k\|_\infty \leq \frac{r_{max}}{1-\gamma}$. We have that:*

$$0 \leq q_* - q_{\pi_k} \leq \left| A_k^1 \frac{E_k}{k} \right| + \frac{4}{(1-\gamma)^2} \frac{r_{max} + \tau \ln |\mathcal{A}|}{k} \mathbf{1},$$

*with $\mathbf{1} \in \mathbb{R}^{\mathcal{S} \times \mathcal{A}}$ the vector whose all components are equal to 1.*

*Proof.* Thanks to Thm. 1, M-VI(1,$\tau$) produces the same sequence of policies that MD-VI($\lambda'$,$\tau'$) with $\lambda' = \tau$ and $\tau' = 0$, and a sequence of q-functions related by $q_k' = q_k - \tau \ln \pi_k$ ($q_k'$ being the q-functions computed by MD-VI($\lambda'$,$\tau'$)). Thm. 1 of Vieillard et al. [32] thus readily applies, the assumption $\|q_k'\|_\infty \leq \frac{r_{max}}{1-\gamma}$ translating into $\|q_k - \tau \ln \pi_k\|_\infty \leq \frac{r_{max}}{1-\gamma}$. $\square$

Notice that that the assumption that $\|q_k - \tau \ln \pi_k\|_\infty \leq \frac{r_{max}}{1-\gamma}$ is not strong, it can be ensured by clipping the $q_k$-values (see also [32, Rk. 1]). Without this, a similar bound would still hold, but with a quadratic dependency of the error term to the horizon, instead of a linear one. Notice that the bound in supremum norm provided in Sec. 3 is a direct corollary of Cor. 1.

Next, we provide a bound for the case $\alpha < 1$.

**Corollary 2.** *Let $(q_k, \pi_k)_{k \geq 0}$ be the sequence of q-functions and policies produced by M-VI($\alpha$,$\tau$), with $\pi_0$ the uniform policy, and with $0 \leq \alpha < 1$. For the sequence of policies $\pi_0, \dots, \pi_k$, we define*

$$P_{k:j} = \begin{cases} P_{\pi_k} P_{\pi_{k-1}} \dots P_{\pi_j} \text{ if } j \leq k, \\ I \text{ else,} \end{cases}$$

*with $I \in \mathbb{R}^{(\mathcal{S} \times \mathcal{A}) \times (\mathcal{S} \times \mathcal{A})}$ the identity matrix. We also define*

$$A_{k:j}^2 = P_{\pi_*(1-\alpha)\tau}^{k-j} + (I - \gamma P_{\pi_{k+1}})^{-1} P_{k:j+1} (I - \gamma P_{\pi_j}), \text{ and } E_k^\alpha = (1-\alpha) \sum_{j=1}^{k} \alpha^{k-j} \epsilon_j.$$

*With these notations, we have*

$$0 \leq q_*^{(1-\alpha)\tau} - q_{\pi_{k+1}}^{(1-\alpha)\tau} \leq \sum_{j=1}^{k} \gamma^{k-j} \left| A_{k:j}^2 E_j^\alpha \right| + \gamma^k (1 + \frac{1-\alpha}{1-\gamma}) \sum_{j=0}^{k} \left( \frac{\alpha}{\gamma} \right)^j \frac{r_{max} + (1-\alpha)\tau \ln |\mathcal{A}|}{1-\gamma} \mathbf{1}.$$

*Proof.* Thanks to Thm. 1, M-VI($\alpha$,$\tau$) produces the same sequence of policies that MD-VI($\lambda'$,$\tau'$) with $\lambda' = \alpha\tau$ and $\tau' = (1-\alpha)\tau$, and a sequence of q-functions related by $q_k' = q_k - \alpha\tau \ln \pi_k$ ($q_k'$ being the q-functions computed by MD-VI($\lambda'$,$\tau'$)). Thm. 2 of Vieillard et al. [32] thus readily applies, with

$$\beta = \frac{\lambda'}{\lambda' + \tau'} = \frac{\alpha\tau}{\alpha\tau + (1-\alpha)\tau} = \alpha,$$

which gives the stated result. $\square$

We refer to [32, Sec. 4.2] for an extensive discussion of this bound, but we highlight the fact that it still shows a compensation of errors (through a moving average instead of the average of Cor. 1), something that is desirable.

## A.4 Details on related works

First, we relate M-VI to CVI. Recall Eq. (3):

$$\begin{cases} \pi_{k+1} = \operatorname{argmax}_{\pi \in \Delta_{\mathcal{A}}^{\mathcal{S}}} \langle \pi, q_k \rangle + \tau \mathcal{H}(\pi) \\ q_{k+1} = r + \alpha \tau \ln \pi_{k+1} + \gamma P \langle \pi_{k+1}, q_k - \tau \ln \pi_{k+1} \rangle + \epsilon_{k+1}. \end{cases}$$

From the Legendre-Fenchel transform, we have that

$$\pi_{k+1} = \operatorname{sm}(\frac{q_k}{\tau}) = \frac{\exp \frac{q_k}{\tau}}{\langle 1, \exp \frac{q_k}{\tau} \rangle} \Leftrightarrow \tau \ln \pi_{k+1} = q_k - \tau \ln \langle 1, \exp \frac{q_k}{\tau} \rangle.$$

Injecting this into the evaluation step, we obtain

$$\begin{aligned} q_{k+1} &= r + \alpha \tau \ln \pi_{k+1} + \gamma P \langle \pi_{k+1}, q_k - \tau \ln \pi_{k+1} \rangle + \epsilon_{k+1} \\ &= r + \alpha(q_k - \tau \ln \langle 1, \exp \frac{q_k}{\tau} \rangle) + \gamma P \langle \pi_{k+1}, q_k - (q_k - \tau \ln \langle 1, \exp \frac{q_k}{\tau} \rangle) \rangle + \epsilon_{k+1} \\ &= r + \gamma P (\tau \ln \langle 1, \exp \frac{q_k}{\tau} \rangle) + \alpha(q_k - \tau \ln \langle 1, \exp \frac{q_k}{\tau} \rangle) + \epsilon_{k+1}, \end{aligned}$$

which is exactly Eq. (5), that is a CVI-like update.

It is a classic result that the sum-log-exp tends towards the hard maximum as the temperature goes to zero (this can be also derived from properties of the Legendre-Fenchel transform):

$$\lim_{\tau \to 0} \tau \ln \sum_a \exp \frac{q_k(s, a)}{\tau} = \max_a q_k(s, a).$$

Using this, the limit of the previous CVI-like update is

$$q_{k+1} = r + \gamma P \langle \pi_{k+1}, q_k \rangle + \alpha(q_k - \langle \pi_{k+1}, q_k \rangle + \epsilon_{k+1}) \text{ with } \pi_{k+1} \in \mathcal{G}(q_k),$$

where we have used that $\max_a q_k(\cdot, a) = \langle \pi_{k+1}, q_k \rangle$ with $\pi_{k+1} \in \mathcal{G}(q_k)$. This is exactly Eq. (6).

## A.5 Proof of Thm. 2

This is indeed a corollary of Thm. 1. First, we handle the case $\alpha < 1$. From Thm. 1, we know that M-VI($\alpha,\tau$) produces the same sequence of policies that MD-VI($\alpha\tau,(1-\alpha)\tau$). From [32, Thm. 2], we now that without error $q'_k = q_k - \alpha \tau \ln \pi_k$ (recall that $q'_k$ is the sequence of $q$-functions computed by MD-VI) converges to $q_*^{(1-\alpha)\tau}$ and that $\pi_k$ converges to $\pi_*^{(1-\alpha)\tau}$ (recall that both algorithms produce the same sequence of policies). From this, we can deduce the limit of $q_k$, the sequence of $q$-function produced by Munchausen VI:

$$\lim_{k \to \infty} q_k = q_*^{(1-\alpha)\tau} + \alpha \tau \ln \pi_*^{(1-\alpha)\tau}.$$

From basic properties of regularized MDPs [15], we know that

$$\pi_*^{(1-\alpha)\tau} = \operatorname{sm}(\frac{q_*^{(1-\alpha)\tau}}{(1-\alpha)\tau}) \Leftrightarrow \ln \pi_*^{(1-\alpha)\tau} = \frac{q_*^{(1-\alpha)\tau}}{(1-\alpha)\tau} - \ln \langle 1, \exp \frac{q_*^{(1-\alpha)\tau}}{(1-\alpha)\tau} \rangle.$$

Therefore, we have that

$$\begin{aligned} \lim_{k \to \infty} q_k &= q_*^{(1-\alpha)\tau} + \alpha \tau \ln \pi_*^{(1-\alpha)\tau} \\ &= q_*^{(1-\alpha)\tau} + \alpha \tau \left( \frac{q_*^{(1-\alpha)\tau}}{(1-\alpha)\tau} - \ln \langle 1, \exp \frac{q_*^{(1-\alpha)\tau}}{(1-\alpha)\tau} \rangle \right) \\ &= \frac{1+\alpha}{1-\alpha} q_*^{(1-\alpha)\tau} - \frac{\alpha\tau}{1-\alpha} \ln \langle 1, \exp \frac{q_*^{(1-\alpha)\tau}}{(1-\alpha)\tau} \rangle. \end{aligned}$$

Noticing that the log-sum-exp does not depend on the actions, we obtain the stated result.

Next, we handle the case $\alpha = 1$. From Thm. 1, we know that M-VI($1,\tau$) produces the same sequence of policies that MD-VI($\tau,0$). From [32, Thm. 1], we now that without error $q'_k = q_k - \alpha \tau \ln \pi_k$

converges to $q_*$ and that $\pi_k$ converges to $\pi_*$, the solutions of the unregularized MDP. To simplify and without much loss of generality, assume that this MDP admits a unique optimal policy. As $q_k = q'_k + \alpha \ln \pi_k$, taking the limit we get for any $s \in \mathcal{S}$

$$\lim_{k \to \infty} q_k(s, a) = \begin{cases} q_*(s, a) \text{ if } \pi_*(a|s) = 1 \\ -\infty \text{ else} \end{cases}.$$

With the adopted convention, this proves the result for the case $\alpha = 1$.

## B   Additional experimental details and results

This appendix provides a complete description of the Munchausen agents, it gives additional experimental details, and it proposes additional results and visualisations:

- Appx. B.1 provides a complete description of the Munchausen agents, as well as some additional details for the considered metrics (such as human scores for games not reported in the literature) and for the learning setting.
- Appx. B.2 discusses the difference between playing $\varepsilon$-greedy and stochastic policies for Munchausen DQN.
- Appx. B.3 discusses the diffrence between using 1-step or 3-steps returns in M-IQN.
- Appx. B.4 provides elements of comparison with the original ALE setting.
- Appx. B.5 provides complementary results for the ablation study.
- Appx. B.6 provides complementary comparison results.

### B.1   Detailed description of the Munchausen agents

All the agents follow a similar learning procedure, described as a pseudo-code in Alg. 1 for M-DQN. What changes is the loss that is optimized.

**M-DQN.**   Here, we recall the basic workings of M-DQN. It estimates a $q$-value through an online $q$-network $q_\theta$ of weights $\theta$. Every $C$ steps, the weights are copied to a *target* network $q_{\bar\theta}$ of weights $\bar\theta$. Transitions $(s_t, a_t, r_t, s_{t+1})$ are stored in fixed-sized FIFO replay buffer. To collect them, M-DQN interacts with the environment using the policy $\mathcal{G}_\varepsilon(\theta)$, the policy that is $\varepsilon$-greedy with respect to $q_\theta$. M-DQN uses (as DQN) a decay on $\varepsilon$ to favour exploration in the beginning of the learning. Each $F$ steps, M-DQN samples a random batch $B$ of transitions from $\mathcal{B}$ and minimizes the following loss, based on the regression target of Eq. (2):

$$\mathcal{L}_{\text{m-dqn}}(\theta) = \tag{7}$$

$$\hat{\mathbb{E}}_B \left[ h\Big(r_t + \alpha \left[\tau \ln \pi_{\bar\theta}(a_t|s_t)\right]_{l_0}^0 + \gamma \sum_{a \in \mathcal{A}} \pi_{\bar\theta}(a|s_{t+1}) \left(q_{\bar\theta}(s_{t+1}, a) - \tau \ln \pi_{\bar\theta}(a|s_{t+1})\right) - q_\theta(s_t, a_t)\Big) \right],$$

with $\pi_{\bar\theta} = \text{sm}(\frac{q_{\bar\theta}}{\tau})$ and $h$ the Huber loss function, with a paremeter $x_h$, $h(x) = x^2$ if $x < x_h$ else $|x|$. A pseudo-code detailing the learning procedure is given in Alg. 1.

**AL.**   We have shown in Sec. 3 that AL can be seen as a limiting case of M-DQN, in the limit $\tau \to 0$. Yet, it cannot be obtained simply by setting $\tau = 0$ in Alg. 1. Instead, we rewrite the minimized loss, according to Sec. 3. Each $F$ steps, AL samples a random batch $B$ of transitions from $\mathcal{B}$ and minimizes the loss

$$\mathcal{L}_{\text{al}}(\theta) = \hat{\mathbb{E}}_B \left[ h\left(r_t + \alpha \left(q_{\bar\theta}(s_t, a_t) - \max_{a \in \mathcal{A}} q_{\bar\theta}(s_t, a)\right) + \max_{a \in \mathcal{A}} q_{\bar\theta}(s_{t+1}, a) - q_\theta(s_t, a_t)\right) \right].$$

**M-IQN.**   IQN is a distributional method. It does not estimate directly a $q$-function, but the distribution of the discounted cumulative rewards, a so-called $z$-function. Precisely, the $z$-function $z_\pi \in \mathbb{R}^{\mathcal{S} \times \mathcal{A}}$ of a policy $\pi$ is a random quantity defined, for each $s, a \in \mathcal{S} \times \mathcal{A}$ as:

$$z_\pi(s, a) = \sum_{t=0}^{\infty} \gamma^t r(s_t, a_t), \text{ with } a_t \sim \pi(\cdot|s_t) \text{ and } s_{t+1} \sim P(\cdot|s_t, a_t) \text{ for } s_0 = s \text{ and } a_0 = a.$$

**Algorithm 1** Munchausen DQN

---

**Require:** $T \in \mathbb{N}^*$ the number of environment steps, $C \in \mathbb{N}^*$ the update period, $F \in \mathbb{N}^*$ the interaction period.
  Initialize $\theta$ at random
  $\mathcal{B} = \{\}$
  $\bar{\theta} = \theta$
  **for** $t = 1$ **to** $T$ **do**
    Collect a transition $b = (s_t, a_t, r_t, s_{t+1})$ from $\mathcal{G}_e(\theta)$
    $\mathcal{B} \leftarrow \mathcal{B} \cup \{b\}$
    **if** $t \mod F == 0$ **then**
      On a random batch of transitions $B_t \subset \mathcal{B}$, update $\theta$ with one step of SGD on $\mathcal{L}_{\text{m-dqn}}$, see (7)
    **end if**
    **if** $k \mod C == 0$ **then**
      $\bar{\theta} \leftarrow \theta$
    **end if**
  **end for**
  **return** $\mathcal{G}_0(\theta)$

---

The $q$-function can be directly related to it with

$$q_\pi(s,a) = \mathbb{E}\left[z_\pi(s,a)\right].$$

A remarkable result is that $z_\pi$ satisfies a Bellman equation, similarly to $q_\pi$, and thus can be estimated with TD. Here, we give a quick overview of IQN, and explain how we modified it. We refer to Dabney et al. [11] for an exact derivation and more details of the original algorithm. IQN estimates the quantile function of $z$ at $\sigma \in [0,1]$, denoted $z_\sigma$. The estimated $q$-value is then $\tilde{q}(s,a) = \mathbb{E}_{\sigma \sim U_{[0,1]}}[z_\sigma(s,a)]$, this expectation being practically approximated by Monte Carlo. The TD error of IQN at step $t$, defined with $\sigma, \sigma' \sim U_{[0,1]}$, is:

$$\text{TD}_{\text{IQN}} = r_t + \gamma z_{\sigma'}(s_{t+1}, \pi(s_{t+1})) - z_\sigma(s_t, a_t), \text{ with } \pi(s) = \underset{a \in \mathcal{A}}{\operatorname{argmax}} \, \tilde{q}(s,a).$$

In practice, $z_{\sigma'}$ is given by a target network, and $z_\sigma$ by an online network, to be optimized. The loss is then estimated as the empirical mean of the TD errors, by sampling $\sigma$ and $\sigma'$ uniformly in $[0,1]$. In M-IQN, we use an additional Munchausen term in TD error,

$$\text{TD}_{\text{M-IQN}} = r_t + \alpha \left[\tau \ln \pi(a_t|s_t)\right]_{l_0}^0 + \gamma \sum_{a \in \mathcal{A}} \pi(a|s_{t+1})(z_{\sigma'}(s_{t+1}, a) - \tau \ln \pi(a|s_{t+1})) - z_\sigma(s_t, a_t)$$

with $\pi(\cdot|s) = \text{sm}(\frac{\tilde{q}(s,\cdot)}{\tau})$ (that is, the policy is softmax with $\tilde{q}$, the quantity with respect to which the original policy of IQN is greedy). We use the same parametrization for $z$ as Dabney et al. [11], and all their provided hyperparameters, as implemented in Dopamine. We used the "Munchausen-RL parameters" from Table 2.

**Custom log-sum-exp trick.** Eq. 7 relies on computing a log-policy, so in our case the log-softmax of a $q$-values. Such computations are usually done using the "log-sum-exp trick", that allows for numerically stable operations by factorizing a maximum. This trick is widely used in software libraries, for example in TensorFlow [1], used to implement the experiments of this work. With this approach, we use the fact that

$$\tau \ln \pi_{k+1} = q_k - \tau \ln \langle 1, \exp \frac{q_k}{\tau} \rangle,$$

that can be unstable if $\tau$ is small. Thus, we compute the log-policy terms using a log-sum-exp-trick as

$$\tau \ln \pi_{k+1} = q_k - v_k - \tau \ln \langle 1, \exp \frac{q_k - v_k}{\tau} \rangle,$$

where we defined $v_k \in \mathbb{R}^{\mathcal{S}}$ as $v_k(s) = \max_a q_k(s,a)$. This is more stable than the one implemented by default, because it takes into account the temperature coefficient.

**Parameters.** We provide the hyperparameters used in our algorithms in Table 2. We denote neural networks structures as follow: $\text{Conv}_{a,b}^{d} c$ is a 2D convolutional layer with $c$ filters of size $a \times b$ and of stride $d$, and $\text{FC}\, n$ is a fully convolutional layer with $n$ neurons. The parameters of the baseline agents are those reported in Dopamine (with the slight modification of considering 1-step returns instead of $n$-step returns for IQN, to match the original paper and the algorithm we modify).

Table 2: Parameters used for Munchausen RL agents.

| Parameter | Value |
|---|---|
| Base (Adam) DQN parameters | |
| $C$ (update period) | 8000 |
| $F$ (interaction period) | 4 |
| $\gamma$ (discount) | 0.99 |
| $|\mathcal{B}|$ (replay buffer size) | $10^6$ |
| $|B_t|$ (batch size) | 32 |
| $e_t$ (random actions rate) | 0.01 (with a linear decay of period $2.5 \cdot 10^5$ steps) |
| $Q$-network structure | $\text{Conv}_{8,8}^{4}\, 32 - \text{Conv}_{4,4}^{2}\, 64 - \text{Conv}_{3,3}^{1}\, 64 - \text{FC}\, 512 - \text{FC}\, n_A$ |
| activations | Relu |
| optimizer | Adam ($lr = 5e - 5$) |
| Munchausen-RL specific parameters | |
| $\tau$ (entropy temperature) | 0.03 |
| $\alpha$ (Munchausen scaling term) | 0.9 |
| $l_0$ (clipping value) | -1 |
| AL specific parameters | |
| $\alpha$ (advantage scaling term) | 0.9 |

**Environment details.** We follow the procedures of Machado et al. [22] to train on the ALE. Notably, we perform one training step (a gradient descent step) every 4 frames encountered in the environment. The state of an agent is the concatenation of the last 4 frames, sub-sampled to a shape of $(84, 84)$, in gray levels. We refer to Machado et al. [22] for details on the preprocessing.

**Metrics.** Here, we recall the definitions of the metrics used to compare algorithms. As an aggregating metric, we use the baseline-normalized score. Every $1M$ frames, we compute the undiscounted return averaged over the last 100 episodes $a_k$, then we normalized it by a random score $r$ and a baseline score $b$ (score after training for 200M steps). The normalized score is then $\frac{a_k - r}{|b - r|}$. We also use human-normalized scores, when we replace the baseline score by the score of a human. We used human scores reported by [23]. For AirRaid, Carnival, ElevatorAction, JourneyEscape, and Pooyan, not considered in Mnih et al. [23], we averaged scores from game-play posted online by players. For a game-per-game metric, we compute the normalized improvement according to a basline. The "final score" of an agent is defined as the score averaged over the last 5M frames. The normalized improvement of a final score $a$ w.r.t. the final score of a baseline $b$ is $\frac{a - b}{|b - r|}$. The maximum scores reported in Table 3 are the maximum scores over training, averaged over 100 episodes, averaged over 3 random seeds, obtained during training.

## B.2 Comparison of greedy and stochastic policies

Although M-DQN naturally produces stochastic policies, we used the $\varepsilon$-greedy one (with respect to $q_\theta$), as explained in Sec. 4. This is motivated by the behaviour of some games. In some games, a random policy fails to gather rewards (as for example Venture or Enduro). The $Q$-network is initialized with small $Q$-values, close to zero. Even with the small temperature $\tau = 0$ we consider, the resulting softmax policy is very close to uniform, and the M-DQN fails to collect rewards, and thus receives no signal to learn. On the converse, an $\varepsilon$-greedy exploration will have a more (randomly) structured exploration, as the scale of $Q$-values does not matter in this case. It then succeed to gather rewards, and to learn something. This is exemplified in Fig. 5, left, for the game Enduro.

Figure 5: Comparsion of M-DQN with a greedy (blue) or stochastic (orange) interaction policy. **Left**: Enduro. **Right:** Seaquest. On Enduro, the stochastic policy is not able to see any reward signal in the beginning, and learns nothing. On Seaquest, we see that it improves over the greedy policy.

Figure 6: Human-normalized scores of M-DQN greedy and stochastic, mean (**left**) and median **right**).

On the converse, if the agent manage to get rewards, the M-DQN agent with a stochastic policy will perform more exploration, and a directed one, as it will chose more often actions with high $Q$-values, thanks to the softmax policy. Consequently, thanks to this less random exploration, it could perform better. We hypothesize that it is what happens for the game Seaquest, shown in Fig. 5, right.

In Fig. 6, we provide the Human-normalized scores of both options, playing with an $\varepsilon$-greedy policy or with the more natural stochastic one. We observe that the stochastic policy is slightly better in median. Yet, it improves less games too, and we kept the $\varepsilon$-greedy policy for the core results. Improving the stochastic policy, maybe with an adaptive temperature or an adaptive $\alpha$ parameter, is an interesting future direction of research.

### B.3 Comparison of $1$-step and $3$-steps learning in M-IQN

The results in the papers are computed with a version of M-IQN that uses 3-steps learning, and compared to version of IQN that also uses 3-steps learning (as it is by default in the Dopamine library). For completeness, we evaluate M-IQN with 1-steps returns, and compre it to IQN with 1-step returns. The human-normalized scores for these algorithms are reported in Fig.7. Theses results show that (1) $n$-step learning and M-RL combine efficiently, as M-IQN 3-steps clearly outperforms M-IQN 1-step and (2) that M-IQN alone (with only 1-step returns) yields already high performances, and it particular outperforms – although by a tight margin – the Rainbow baseline, that uses 3-steps returns.

### B.4 Element of comparison with the original ALE setting

We explained in Sec. 4 the difference between the ALE setting we consider, more modern and more difficult, compared to the ALE setting often considered, for example for the seminal DQN [23] or for Rainbow [18]. The Rainbow baseline we consider [10] is also not exactly the published one: even if the most important features are included, as deemed by Hessel et al. [18], it does not include all features (such as double $Q$-learning or dueling architecture).

Figure 7: Human normalized scores of M-IQN, IQN, and Rainbow with different $n$-steps returns, mean (**left**) and median **right**). M-IQN, IQN, and Rainbow use 3-steps, while the other two use 1-step.

As a (partial) check, we also evaluated our Munchausen agents, M-DQN and M-IQN, as well as the baselines DQN, IQN and Rainbow, in a setting as close as possible to the one used for the baselines' publications. Notably, here we did not used sticky actions, making the environment deterministic, and we end an episode whenever the agent loses a life, instead of when it encounters a game-over. We also use hyperparameters provided in the original publications, the only difference being that we used a target update period of $10000$ steps instead of $8000$. We did so on the Asterix game, the results being depicted in Fig. 8.

On Fig. 8, left, we can observe DQN and M-DQN. The result for DQN is normal, despite the apparent "crash", see for example the training curves in [18] (notice also that it is often the best scores over training which is reported, instead of the final one, as in our Tab. 3 or in the seminal DQN publication [23]). We can observe that M-DQN performs much better than DQN, without falling, and that the score is close to the one of M-DQN in the more difficult setting (15k vs 19k in the more difficult setting).

On Fig. 8, right, we can observe Rainbow, IQN and M-IQN. All algorithms perform pretty well. For example, Rainbows reaches roughly 350k, comparable to the original publication[7]. This is much more than in our setting, where Rainbow reaches only 18k, suggesting that the original setting is easier. We can also see that IQN works well (and somehow surprisingly better than in the original publication, compared to Rainbow), and that M-IQN works better than both IQN and Rainbow.

An interesting thing is to see how the methods degrades (roughly) when going from the agent is trained in the considered setting, compared to the original one. Rainbow goes from 350k to 18k (5% of the original scores), IQN goes from 350k to 33k (10%), while M-DQN goes from 15k to 17k (113%) and M-IQN goes from 350k to 50k (17%). This suggests that M-RL might be more stable over environments.

For sure, this discussion only holds for one game, and no general conclusion can be drawn. Yet, it suggests a few things, the ALE setting we consider is more difficult, among other advantages [22], the Rainbow baseline we consider is correct, and M-RL seems to be more stable.

## B.5 Additional results on the ablation study

We provide complementary results regarding the ablation study:

- Fig. 9 p. 22 reports the Rainbow-normalized scores of the ablation (instead of the Human-normalized ones in the main paper, Fig. 3).

- Fig. 11 p. 24 shows the normalized improvements of all ablations with respect to DQN.

- Fig. 12 p. 25 reports all learning curves an the 60 Atari games for the ablation.

Figure 8: Scores of different agent on the game Asetrix, using the original ALE. **left:** M-DQN and DQN. **right:** Rainbow, IQN and M-IQN.

Figure 9: Rainbow-normalized ablation study results. **Left:** mean. **Right:** median.

The Rainbow-normalized scores (Fig. 9) confirms the Human-normalized ones (Fig. 3). The scores themselves are different (due to a different normalization), but the order of the different variations and their gaps is comparable.

Fig. 11 provides a summary of the per-game improvement, while Fig. 12 provides all related learning curves (Fig. 11 summarizing what the results are after 200M frame). We can observe that M-DQN is not always the best performing agent. Yet, it is very often competitive with the best performing ablation (when M-DQN does not perform the best), and the ablation that surpasses M-DQN is highly game-dependent. Overall, M-DQN is consistently the best performing agent over the whole suite of games, as confirmed by Fig. 3 or Fig. 9 both in mean and median Rainbow and Human-normalized scores.

AL performs pretty well (even if less well than M-DQN). Yet, Munchausen-RL is more general, as it consists only in adding a scaled log-policy term to the reward. We've shown in the main paper how it can be readily applied to agents that does not even consider stochastic policies. On the converse, ALE relies heavily on being able to compute the maximum $Q$-value, something which could not be easily extended to continuous actions, contrary to the Munchausen principle. We let this as an interesting direction for future work. In both average and mean (Fig. 3 and 9), Soft-DQN is the worst ablation, despite being much better in a few games (for example, Amidar or Jamesbond). Again, the temperature was not specifically tuned for Soft-DQN, but it is on par with the close literature (see discussion in Sec. 4). This suggests that the maximum entropy RL principle alone might not be sufficient, especially when one observes the significant improvement that the Munchausen term brings to it (or, implicitly, adding KL regularization to the entropy term). We also notice again that Adam DQN works surprisingly well, compared to the original DQN. This is a very interesting finding, and it suggests that Adam DQN should be considered as a better baseline than the seminal DQN.

Figure 10: Rainbow-normalized scores. **Left:** mean. **Right:** median.

## B.6 Additional comparison results

For completeness, we provides additional comparison results:

- In addition to the Human-normalized results of Fig. 1, we provide a Rainbow-normalized comparison of the Munchausen agents with respect to DQN, C51, IQN and Rainbow in Fig. 1.
- In addition to the per-game normalized improvement of a Munchausen agent with respect to its natural baseline (Fig. 4), we provide the per-game improvement for M-DQN over DQN, C51, IQN and Rainbow in Fig. 13, as well as the per-game improvement of M-IQN over the same baselines in Fig. 14.
- We provide a summary of all best scores (among training, averaged over 3 seeds), for all games on all agents, in Table 3 p. 28. M-IQN obtains the most highest-ranking scores among all the considered baselines (including the human one).
- For completeness, we report all learning curves of the Munchausen agents and the considered baselines, for the full set of Atari games, in Fig. 15.

These additional results confirm the observations made in the main paper.

Figure 11: Per games N.I./DQN of the ablation study.

Figure 12: All averaged training scores of the ablation. M-DQN in blue, AL in orange, Soft-DQN in green, DQN Adam in red, and DQN in dashed purple.

Figure 13: Normalized Improvement of M-DQN vs DQN, C51, IQN, and Rainbow.

Figure 14: Normalized Improvement of M-IQN vs DQN, C51, IQN, and Rainbow.

Table 3: Maximum scores obtained during training (averaged over 100 episodes and 3 random seeds). The bottom line counts the number of games on which an algorithm or a human performs the best.

| | random | human | IQN | DQN | RAINBOW | M-DQN | M-IQN |
|---|---|---|---|---|---|---|---|
| AirRaid | 400 | 3000 | 15077 | 7700 | 14056 | 8914 | **19111** |
| Alien | 228 | **7128** | 5119 | 2533 | 3587 | 3795 | 4492 |
| Amidar | 6 | 1720 | 2442 | 1222 | **2630** | 1423 | 1875 |
| Assault | 222 | 742 | 4902 | 1573 | 3511 | 2165 | **7504** |
| Asterix | 210 | 8503 | 10965 | 3433 | 18367 | 17238 | **49865** |
| Asteroids | 719 | **47389** | 1616 | 828 | 1489 | 1150 | 1685 |
| Atlantis | 12850 | 29028 | 893764 | 919622 | 838590 | **939533** | 918183 |
| BankHeist | 14 | 753 | 1073 | 704 | 1148 | 1190 | **1292** |
| BattleZone | 2360 | 37188 | 41475 | 18667 | 40895 | 36509 | **52517** |
| BeamRider | 364 | **16926** | 7365 | 5852 | 6529 | 6745 | 12775 |
| Berzerk | 124 | **2630** | 662 | 559 | 842 | 608 | 736 |
| Bowling | 23 | **161** | 46 | 33 | 49 | 37 | 32 |
| Boxing | 0 | 12 | 98 | 82 | 99 | 98 | **99** |
| Breakout | 2 | 30 | 159 | 127 | 120 | **331** | 320 |
| Carnival | 380 | 4000 | **5712** | 4860 | 5069 | 5022 | 5588 |
| Centipede | 2091 | **12017** | 3816 | 3337 | 6618 | 4134 | 4371 |
| ChopperCommand | 811 | 7388 | 9301 | 2852 | **12844** | 4507 | 4573 |
| CrazyClimber | 10780 | 35829 | 137201 | 109635 | 147743 | 140156 | **150783** |
| DemonAttack | 152 | 1971 | 15433 | 6411 | 17802 | 12114 | **68825** |
| DoubleDunk | -19 | -16 | 21 | -6 | **22** | 0 | 22 |
| ElevatorAction | 0 | 3000 | 67224 | 1723 | 79968 | 4215 | **89237** |
| Enduro | 0 | 860 | 2270 | 815 | 2230 | 1643 | **2332** |
| FishingDerby | -92 | -39 | 45 | 9 | 43 | 44 | **55** |
| Freeway | 0 | 30 | 34 | 26 | 34 | 34 | **34** |
| Frostbite | 65 | 4335 | 8061 | 1186 | 8572 | 5453 | **9538** |
| Gopher | 258 | 2412 | 12108 | 6044 | 10641 | 14728 | **27469** |
| Gravitar | 173 | **3351** | 1350 | 330 | 1272 | 550 | 1134 |
| Hero | 1027 | 30826 | 36583 | 17330 | **46764** | 13824 | 26037 |
| IceHockey | -11 | 1 | -0 | -6 | 2 | 0 | **12** |
| Jamesbond | 29 | 303 | **3596** | 589 | 1106 | 814 | 1637 |
| JourneyEscape | -18000 | -1000 | -1252 | -2668 | -959 | -938 | **-806** |
| Kangaroo | 52 | 3035 | 12872 | 12192 | 13460 | **14067** | 10939 |
| Krull | 1598 | 2666 | 8910 | 6410 | 6229 | 8912 | **10703** |
| KungFuMaster | 258 | 22736 | **33348** | 24495 | 27900 | 29607 | 27119 |
| MontezumaRevenge | 0 | **4753** | 500 | 2 | 500 | 0 | 0 |
| MsPacman | 307 | **6952** | 5225 | 3471 | 4027 | 4544 | 6029 |
| NameThisGame | 2292 | 8049 | 9129 | 7348 | 9229 | 11807 | **12761** |
| Phoenix | 761 | 7243 | 5137 | 5651 | **8605** | 5140 | 5327 |
| Pitfall | -229 | **6464** | -3 | -17 | -1 | 0 | 0 |
| Pong | -21 | 15 | **20** | 17 | 20 | 19 | 19 |
| Pooyan | 500 | 1000 | 5339 | 3535 | 5640 | 6396 | **13096** |
| PrivateEye | 25 | **69571** | 6852 | 1004 | 21532 | 121 | 100 |
| Qbert | 164 | 13455 | 16995 | 10399 | **18503** | 16415 | 14739 |
| Riverraid | 1338 | 17118 | 15554 | 12051 | **21091** | 19346 | 16271 |
| RoadRunner | 12 | 7845 | 59443 | 39468 | 55300 | 51866 | **61269** |
| Robotank | 2 | 12 | 67 | 61 | 66 | 66 | **73** |
| Seaquest | 68 | **42055** | 19170 | 2133 | 11362 | 2666 | 23885 |
| Skiing | -17098 | **-4337** | -11035 | -15712 | -20518 | -9671 | -10336 |
| Solaris | 1236 | **12327** | 2204 | 1955 | 2438 | 5169 | 5765 |
| SpaceInvaders | 148 | 1669 | 5452 | 1850 | 4420 | 7504 | **13871** |
| StarGunner | 664 | 10250 | **80362** | 45015 | 57909 | 55100 | 65757 |
| Tennis | -24 | -8 | **23** | -0 | 0 | 0 | 0 |
| TimePilot | 3568 | 5229 | 11887 | 3768 | 12283 | 10590 | **15155** |
| Tutankham | 11 | 168 | **256** | 132 | 245 | 200 | 207 |
| UpNDown | 533 | 11693 | 74659 | 10348 | 39065 | 45738 | **216080** |
| Venture | 0 | 1188 | 1430 | 52 | **1579** | 19 | 1101 |
| VideoPinball | 0 | 17668 | 485551 | 177488 | 513484 | 368930 | **625118** |
| WizardOfWor | 564 | 4756 | 6208 | 2597 | 8201 | 12517 | **13644** |
| YarsRevenge | 3093 | 54577 | 85762 | 24389 | 45567 | 29792 | **111583** |
| Zaxxon | 32 | 9173 | 11761 | 4825 | 15089 | 13905 | **19080** |
| Best | 0 | 14 | 7 | 0 | 8 | 3 | **28** |

Figure 15: All averaged training scores. M-DQN in blue, M-IQN in orange, IQN in dashed green, Rainbow in dashed red, DQN in dashed purple, and C51 in dashed brown.