[Reviews · NeurIPS 2020]

Review 1

Summary and Contributions: This paper introduces Munchausen Reinforcement Learning, a novel approach to bootstrapping in RL. The approach is elegant and can easily be incorporated into RL algorithms based on the TD scheme. The idea is to augment the environment reward with a scaled logarithm of the agent's policy. The authors incorporate this idea into DQN and IQN. The addition to DQN involves (i) changing DQN's regression target to that of Soft DQN (or a simple discrete-action SAC) and (ii) augmenting the reward with the scaled log-policy. The theoretical analysis of M-RL (established through theoretical analysis based on M-VI) show that (i) it performs implicit KL regularization and (ii) it increases the action-gap (in a quantifiable fashion).

Strengths: - The paper is a pleasure to read; simple to follow due to its great organization, clarity, and consistency! - The paper is exciting from both theoretical and empirical sides. - The idea is super elegant; when applied to DQN, it does not require more than a simple change to the regression target. - The results on the Atari 2600 games (all 60 games of ALE) are very impressive! (a) When incorporated into the (vanilla) DQN, M-DQN significantly outperforms DQN and C51, and is competitive with Rainbow! (b) When combined with IQN, M-IQN significantly outperforms Rainbow and IQN (setting a new SOTA).

Weaknesses: A missing discussion w.r.t. the findings of (van Seijen et al., 2019) regarding the action-gap (which concluded that homogeneity of the action-gap across the state space to be the actual effect needed, and not increasing it). Following this, Log-DQN would've been the right candidate for the study of the action-gap (i.e., Fig. 2). Especially, given that M-DQN is showing a more homogenous action-gap across timesteps versus AL (which increases the action-gap but doesn't achieve homogeneity). If the action-gaps of Log-DQN and M-DQN show similar homogeneity (as expected from Log-DQN), then it could hint at a potentially different property of M-RL: M-RL could be implicitly achieving more homogeneous action-gaps across the state space in effect. The finding that action-gaps are increased by (1+alpha)/(1-alpha) for M-VI is not necessarily proof to the contrary of this effect, as action-gaps could be scaled due to other regularizing effects in M-RL. This would be a very valuable addition, as it could also explain the performance gap between AL and M-DQN in the sense of their effect on the action-gap. Additionally, it would put the analyses of this paper in regards to action-gap in perspective w.r.t. the latest findings from the literature. (P.S. there could be a hidden connection between the logarithmic mapping in Log-RL and the log(pi) additive reward in M-RL). If a discussion on possible connections to Log-RL is added and a comparison to Log-DQN is added in the action-gap study of Fig. 2, I'd be more than happy to increase my score. This is the only gap I see in this, otherwise, superb paper!

Correctness: All is well-done. Only I suspect the discussions on increasing the action-gap (as discussed in the "Weaknesses" section).

Clarity: Excellent - a joy to read!

Relation to Prior Work: As discussed as the "Weaknesses", discussions on possible connections to Log-RL are missing.

Reproducibility: Yes

Additional Feedback: >>> After Authors' Reponse: I still find the paper's analysis regarding action-gaps a bit weak, and the authors' response didn't help much in that regard. I think their action-gap analysis needs to be considered under the new findings of (van Seijen et al., 2019); increasing the action-gap is not important on its own, rather it's the homogeneity of the action-gaps across the states that is important. While I still stand by my verdict of accepting this paper, in light of other reviews, I think the paper's writing should be toned down a bit in regards to its theoretical novelty and claims about empirical results (e.g. the first non-dist-RL to beat a dist-RL). >>> Qs. Q1: To the best of my knowledge, IQN in Dopamine also uses Double Q-learning. Is this also the case for your M-IQN agent? If yes, I believe it has not been mentioned. Also, it would mean that your M-IQN agent uses also Double Q-learning beyond your Rainbow baseline (which doesn't use Double Q-learning in its Dopamine implementation). Q2: Couldn't the improvement of M-DQN over AL be partly due to M-DQN achieving a more homogeneous action-gap across the state-space as opposed to AL which increases the action-gap but doesn't achieve homogeneous action-gap sizes? If so, this would conform to the findings of (van Seijen et al., 2019). To study this, I believe that Log-DQN (from the latter work) should also be included in your action-gap study of Fig. 2 (i.e. compare against its action-gap on Asterix). Q3: Can you comment on any possible connections between Log-RL (van Seijen et al., 2019) and M-RL? Does the log(pi) of the additive reward in M-RL relate to the log(q) in Log-RL in any way? Q4: In Fig. 1, are all agents using the Adam optimizer except for DQN? From your ablation study, it seems to me that this is the case. Doesn't this make these results a lot more moderate in the case of M-DQN (given that the competitiveness of M-DQN with Rainbow becomes less impressive once DQN-Adam is also shown)? I think this should be clearly stated right there when discussing the results of Fig. 1 or, alternatively, DQN-Adam curves can be added to clearly illustrate this. Q5: Doesn't AL improve beyond C51? If it does, since AL doesn't use distRL (if I'm not mistaken), doesn't it make M-DQN the *second* known non-distRL agent to beat C51? (To understand where my confusions come from: I'm finding it difficult to put the performances of Fig. 3 in perspective w.r.t. those of Fig. 1 -- have to compare them back and forth to sanity checking things for myself). >>> Refs. (van Seijen et al., 2019) Using a Logarithmic Mapping to Enable Lower Discount Factors in Reinforcement Learning.


Review 2

Summary and Contributions: This paper proposes to add a term of a scaled log probability of an action to an immediate reward for soft Q-learning. The proposed method is theoretically justified as KL regularization and related to conservative policy iteration, dynamic policy programming, and advantage learning. The proposed method is empirically evaluated on 60 Atari games, achieving good performance. It is shown that DQN with soft Q-learning, Adam, and the Munchausen term alone is competitive with Rainbow and that IQN with soft Q-learning and the Munchausen term outperform Dopamin's Rainbow and IQN.

Strengths: The proposed method is relatively simple but surprisingly effective, achieving significantly better scores than both Dopamin's Rainbow and IQN, which alone is a significant contribution to deep RL research.

Weaknesses: The proposed method has three additional hyperparameters, which could be a significant drawback in practice. This paper does not provide any empirical analysis on how those hyperparameters matter. Only a single set of values used in the experiments is provided. Therefore, it is unclear how difficult it is to tune these values in practice. Having such an analysis would enhance the paper a lot. The 60-game Atari results are not comparable to the published results in the Rainbow and IQN papers because of sticky actions and life-loss episode termination. Dopamin's Rainbow is also different from the one proposed by the Rainbow paper. Therefore, it is still inconclusive how well M-DQN and M-IQN would perform relatively to Rainbow and DQN on their original settings.

Correctness: M-IQN is said to install a "new state of the art" of non-distributed RL, but beating IQN and Rainbow alone is not enough to say so because FQF (http://papers.neurips.cc/paper/8850-fully-parameterized-quantile-function-for-distributional-reinforcement-learning) outperformed Rainbow and IQN in terms of both mean and median human normalized scores. L48-49 says M-RL "is the first one that allows quantifying this increase", which implies that the gap-increasing operators proposed by Bellemare et al. do not allow quantifying the increase of the action gap, but I cannot find the explanation of why.

Clarity: It is well written except a few typos. L189 MPD -> MDP L654 ALE -> AL

Relation to Prior Work: Its relation to DPP, AL, and soft Q-learning are clearly discussed.

Reproducibility: Yes

Additional Feedback: Post rebuttal: In the rebuttal, the authors mentioned a plan to add a hyperparameter sensitivity analysis and M-FQF results, which could partly address the weakness of the paper. I keep my score of 7.


Review 3

Summary and Contributions: The paper presents an algorithm that the authors call Munchausen reinforcement learning (M-RL) which applies an entropy regularizer to the reward signal. The authors show that applying this regularization to the well-known DQN algorithm results in strong performance in the Atari benchmark suite.

Strengths: The authors show that the proposed algorithm works very well in the Atari benchmark suite, reaching a level of performance comparable to the state-of-the-art.

Weaknesses: M-RL and its variants are essentially instances of known algorithms for entropy regularized reinforcement learning, as the authors themselves point out several times in the text. Hence there is a big question mark for novelty, and I find it questionable to give a new name (M-RL) to this family of known algorithms.

Correctness: I do not agree with the claim that the proposed algorithm is novel. Most results appear correct, potentially with one exception (see my comments below).

Clarity: Yes, I believe the paper is well written.

Relation to Prior Work: The authors do make an effort to relate the proposed algorithm to existing algorithms, but there is no strong evidence that the proposed algorithm is novel. Regularizing the reward on the log policy (which the authors apparently believe is a novel contribution compared to Soft-DQN) is discussed at length in the following paper: Neu et al. (2017) A unified view of entropy-regularized Markov decision processes

Reproducibility: Yes

Additional Feedback: Most theoretical results presented in the paper are previously known, including the forms of regularized value iteration and its relation to mirror descent. The only attempt at a novel theoretical result is that at the bottom of page 4, where the authors claim that the bound on the approximation error for exact updates applies to their deep learning setting. I find this hard to believe, especially since Vieillard et al. explicitly say that their analysis does not apply to deep learning. The motivation for this result is given in a single sentence, saying that implicit KL regularization does not introduce errors in the greedy policy. If the result is indeed true, then I think the authors have to spend considerably more effort explaining why. The concept of log-policy clipping is also common in entropy regularized RL, since this is the only way to guarantee numerical stability. Post-rebuttal: Evidently I do not agree that I have "deeply misunderstood the contribution", though I may have been a bit harsh in my judgment. To make my points more clear, I will attempt to describe my perspective better. What I don't like about the paper (and a major reason for my lower score) is the claim that the algorithm is clearly different from entropy regularization (granted, it appears that the authors' definition of entropy regularization is significantly more narrow than mine). In fact, there is a strong relationship to several previous algorithms: - The regularized Bellman optimality equation can be written as follows (in simplified notation): V(s) = \max_a [ r(s,a) - \beta \log \pi(s,a) + \sum_t p(s,a,t) V(t) ] It so happens that the max operator simplifies to the log-sum-exp operator for softmax policies, but you might as well perform a TD style update in the way that you do. This is the reason for my claim that M-DQN is similar to soft Q-learning in spite of the different update rules: the two are based on different versions of the same Bellman equation. - I'm not convinced that changing the sign of the regularization factor has a significant impact on the theoretical results. As far as I can tell, the analysis of previous algorithms still holds even if you allow the regularization factor to have a negative value. - There is also a strong relationship to a series of algorithms that perform dual regularization using the KL divergence from the previous policy *and* the entropy of the policy (just as in your result from Theorem 1): Abdulsamad et al. (2017) State-Regularized Policy Search for Linearized Dynamical Systems Proceedings of ICAPS Pajarinen et al. (2019) Compatible natural gradient policy search Machine Learning The main difference is that those earlier algorithms start from a policy gradient perspective and derive a Bellman equation, while you do the opposite (start from a Bellman update and derive a policy gradient rule). Now, admittedly, it does seem as if the update rule of M-DQN leads to novel theoretical results. Specifically, the double regularization in the update rule appears equivalent to proximal + global regularization, even though *none of the regularization terms is proximal*. In my opinion, this theoretical result should be the main claim of the paper. Regarding the result on the bottom of page 4, I had a look at the supplementary material. Your proof relies on that of Vieillard et al., and you state that your algorithm and that of Vieillard et al. produces exactly the same sequence of value functions and policies. Yet Vieillard et al. explicitly state that their result does *not* apply to deep RL, but you claim that it does for your algorithm. This seems like a contradiction, and if there is some difference in how the algorithms progress for parameterized value functions and policies, this should be much more clearly explained (in my case, the part about not introducing errors in the greedy step does not provide enough detail).


Review 4

Summary and Contributions: This paper proposed a simple idea to modify generic TD algorithms by a (scaled) log of policy to the reward. The idea was applied to DQN and Implicit Quantile Network (IQN) and it outperforms Rainbow on Atari games. It sounds like an ad-hoc modification, but they showed that this is equivalent to applying a Kullback-Leibler regularization and increase action gap.

Strengths: The performance of the algorithm is surprisingly well. The paper showed that it is equivalent to add a regularization to successive policies. This is an interesting observation (and Theorem 1 is a similar one for the case of Vieillard's M-VI algorithm). Theorem 2 shows that the M-VI algorithm increases the limit (as iteration increases to infinity) of the difference between the greedy action and (all) the q values is a (1+alpha)/(1-alpha) bigger than the optimal policy, where alpha is the scaling factor of the log policy.

Weaknesses: (1)The paper claimed this idea can work with any TD algorithm. can you try in tabular lookup table or linear case? to better understand your idea? I feel the performance on Atari may be due to the fact that this "reward signal" encourages deterministic policies and they happen to work well on the problems. Atari performance is great. However, it is hard to dig into why one performs well and bad (see the Montezuma revenge question below). (2)A comparision with soft DQN (eqn 1; which is discrete-action version of soft-actor-critic) is missing. so it's not very clear if the addition of the log policy helps. The algorithm was developed by modifying soft DQN, but didn't compare with it. so a natural question is did the performance come from softDQN element (the blue part in eqn 1) or the log of the policy (which is your idea)? (3) Th. 2 may be the reason for the outstanding performance of the algorithm. Fig 2 shows the action gap together with Advantage Learning. Advantage Learning (AL) has a bigger action gap. I'm not sure in the performance studies AL was compared fairly. line 258: "Both M-DQN and AL increase the action-gaps compared to Adam DQN. If AL increases it more, it seems also to be less stable". It is true for your case too, when alpha gets close to 1, your algorithm also becomes unstable. After rebuttal: I've read the other reviews. I tend to interpret this paper is towards more empirical one because Th 1 is kind of straightforward and not surprising. Theorem 2 is more interesting, but the increasing action gap is only shown empirically effective in previous work. In fact, only the order of actions matters. It is hard for me to justify the value of this result. About comparison to Soft DQN: since it's called the "ablation" studies, the parameter used for Soft DQN is the same for your algorithm (complete elements). I feel this may not be the best case for Soft DQN. About algorithm novelty: I think the algorithm is reasonably novel. The algorithm adding log policy to the reward to modify Soft DQN this way essentially used TD algorithm on the policy as well. I don't think it's worthwhile for the authors arguing about a strong novelty here, instead, maybe clearly identify the benefits of bringing more action gap and do more comparative studies to Advantage and Soft-DQN algorithms. The rebuttal shows some change in empirical results with advantage of the new algorithm being smaller than in the paper. You may need to spend more time in experiments to make sure. I tend to encourage the authors to spend more time in the above aspects and improve the paper much better. This is a good paper and deserve a better and more investigation.

Correctness: yes

Clarity: yes

Relation to Prior Work: yes

Reproducibility: Yes

Additional Feedback: MonteZumaRevent is at the two ends of the algorithms. do you know why? M-DQN: your algorithm name was not defined. eqn below line 82: This is slightly different from the DQN. eqn 1: what is sm()? later I found it. It is bit hidden. By adding this kind of pseudo reward signal, it is noticeable so here you have a "bootstrap" too for the policy too.

[Author Response · NeurIPS 2020]



Figure 1: Left to right: all mean, all median, ablation mean, ablation median.

**General.** Shortly after submission, we found an unintentional inconsistency in our evaluation protocol. Our M-Agents
and their ablations were evaluated in a slightly different Atari setting from the others (episode terminated on life loss
rather than game over), favoring them on some games. We reran experiments with the correct settings and (human-
normalized) results are on Fig. 1. Despite some drop in absolute performance (eg, M-DQN outperforms C51 but is a bit
less competitive with Rainbow), it does not change the ranking of algorithms (for all metrics) nor our conclusions.

**R1. Q1:** In Dopamine, IQN doesn't use double Q-learning (`implicit_quantile_agent.py`, ll. 192-196). **Q2:**
Studying the homogeneity of the action gap is an interesting research direction that we didn't investigate, thanks for
pointing it out. In the limit ($\alpha = 1$), it is homogeneous (because infinite), but with numerical instabilities. We'll
add a comparison with log-DQN in Fig. 2, to get at least some empirical insights. **Q3:** We do not see any obvious
connections between Log-RL (related to the $h$-transform of Pohlen *et al.*) and M-RL (justified here through the lens of
KL regularization), but that's also an interesting research direction. We also think that, even if different, both approaches
could be combined to build an even stronger agent. **Q4:** Yes, exactly, we say why in footnote 6 (and this comparison is
done in the ablation). We'll say it earlier. **Q5:** AL is indeed better than C51 on some metrics. Yet, this was not observed
in [6], probably because the authors use RMS (to compare with the standard DQN) while we use Adam (to compare
with M-DQN). Additionally, note that AL is a special case of M-DQN. We will rephrase appropriately.

**R2. Parameters:** To choose the parameters, we did a sensitivity analysis on a subset of games, that we'll add to the
Appx. Yet, note that the parameters were not selected through a simple grid search. Values of $\tau$ are on par with the ones
provided by the analysis in [30], and $\alpha$ is the same as the one found optimal in [6]. In terms of "easiness to tune", our
empirical findings suggest that the most sensitive parameter is $\tau$, while it is rather easy to find working values for $\alpha$ and
$l_0$. **ALE:** our results are indeed not comparable with the original DQN ones, as discussed ll.217-231. Unfortunately, the
authors used a proprietary version of ALE, so exact comparison to their results is not possible. **FQF:** thanks, we missed
this paper. However, FQF does not use sticky actions and comparison is thus not straightforward. Our method readily
applies to FQF (as for IQN), and we will try to add M-FQF results in the paper. If it is not possible, we will reformulate
to soften the state-of-the-art claim. We believe this does not hinder the relevance of our approach, as M-IQN still
outperforms Rainbow. **Action-gap:** by "quantifying", we mean "analytically quantifying": Bellemare et al. show that
the action gap increases but not by how much, while we derive an actual value for the increase (Thm. 2).

**R4.** We trust that R4 has deeply misunderstood our contribution. Indeed, they state that "M-RL applies an entropy
regularizer to the reward signal", and all the following (mostly negative) comments rely heavily on this statement. This
is just wrong and we start the paper by stating otherwise (l.21-24, " We insist right away that this is different from
maxent RL, that *subtracts* the scaled log-policy to *all* rewards [...]" while we *add* it to the *immediate* reward). **On
novelty:** We strongly disagree, our contribution is not simply an instance of entropy-reg RL, as notably thoroughly
discussed in Sec. 3 and related Appx. M-DQN *is different* from Soft-DQN, one just has to compare Eqs. (2) and (3)
(paying attention to the *signs* of the different log terms). We're also absolutely certain that neither our algorithms nor
our analysis are covered by Neu et al. **Most theoretical results are previously known.** Again, we strongly disagree.
We're very clear about what our contributions are, and Thm. 1 and 2 are new (we could reevaluate this claim if a ref was
provided). **Hard to believe** (about the theoretical result): we don't ask to believe our claim, as we provide the proofs,
see Appx A.2 and Cor. 1 and 2 in Appx. A.3. Shortly, the implicit KL regularization avoids the error in the greedy step,
that cannot be avoided when the KL regularization is explicit (the only case considered in [30]). **Clipping:** Obviously,
we do not claim to be the first to clip a log term, but we provide this kind of details for the sake of reproducibility.

**R5. Linear case:** in this setting M-VI is strictly equivalent to MD-VI (see Thm. 1, noticing that in the linear case there
is no error in the greedy step), so we refer to [30] for a study on tabular MDPs. Notice that this equivalence is lost with
a non-linear paremeterization (due to the necessary error in the greedy step with explicit KL regularization), hence the
interest of the Munchausen principle for *deep* RL. **Comparison to Soft-DQN is missing:** no, it's not, it is provided in
Fig. 3 (ablation). **Unsure about AL fairly compared**: we use $\alpha = 0.9$ for both M-DQN and AL, and this choice is
consistent with the AL paper, so we think the comparison to be fair. Also, notice that AL is a special case of M-DQN.
**Montezuma**: this is a hard exploration game, and our method is not designed to favour exploration (there is no signal
to reinforce, a discussion to a related issue is provided in Appx. B.2). **M-DQN** is defined l.31. **Eq. l.82:** this is exactly
DQN, as $\pi$ is defined to be greedy wrt $q_{\bar{\theta}}$. **Bootstrap:** yes, we totally agree with this, it is actually the reason for the
name "Munchausen" (l.18-26).

[Meta-Review · NeurIPS 2020]

In this submission, a new bootstrapping optimization technique is proposed, based on the idea of adding the log-policy to the immediate reward. This is shown to bring strong empirical gains, and the theoretical analysis helps understand why. Although reviewers remained divided even after an active discussion period (7, 7, 5, 5), I believe this is a paper worth publishing at NeurIPS. Simple ideas bringing significant improvements, like this one, are typically those most impactful. I also appreciate the efforts made to better understand the theoretical properties of the proposed algorithm, beyond the basic intuition. The main remaining concerns of R4 and R5 were related to the significance of the theoretical results. However, in my opinion they did not provide specific enough criticism that could confidently invalidate these results, nor disprove their novelty. Consequently, I am recommending acceptance, aligning myself with R1 and R2.